# Markerless tracking of an entire honey bee colony

Katarzyna Bozek [1,5,✉], Laetitia Hebert [1], Yoann Portugal[1,2], Alexander S. Mikheyev[2,3,6], & Greg J. Stephens [1,4,6],

From cells in tissue, to bird flocks, to human crowds, living systems display a stunning variety of collective behaviors. Yet quantifying such phenomena first requires tracking a significant fraction of the group members in natural conditions, a substantial and ongoing challenge. We present a comprehensive, computational method for tracking an entire colony of the honey bee *Apis mellifera* using high-resolution video on a natural honeycomb background. We adapt a convolutional neural network (CNN) segmentation architecture to automatically identify bee and brood cell positions, body orientations and within-cell states. We achieve high accuracy (~10% body width error in position, ~10° error in orientation, and true positive rate > 90%) and demonstrate months-long monitoring of sociometric colony fluctuations. These fluctuations include ~24 h cycles in the counted detections, negative correlation between bee and brood, and nightly enhancement of bees inside comb cells. We combine detected positions with visual features of organism-centered images to track individuals over time and through challenging occluding events, recovering ~79% of bee trajectories from five observation hives over 5 min timespans. The trajectories reveal important individual behaviors, including waggle dances and crawling inside comb cells. Our results provide opportunities for the quantitative study of collective bee behavior and for advancing tracking techniques of crowded systems.

[1] Biological Physics Theory Unit, OIST Graduate University, Okinawa, Japan. [2] Ecology and Evolution Unit, OIST Graduate University, Okinawa, Japan. [3] Research School of Biology, Australian National University, Canberra, Australia. [4] Department of Physics and Astronomy, Vrije Universiteit Amsterdam, Amsterdam, The Netherlands. [5] Present address: University of Cologne, Center for Molecular Medicine Cologne (CMMC), Faculty of Medicine and University Hospital Cologne, Cologne, Germany. [6] These authors contributed equally: Alexander S. Mikheyev, Greg J. Stephens. ✉email: k.bozek@uni-koeln.de

Among the rich phenomenology of organismal behavior, honey bees and other eusocial animals are distinguished by their remarkable, self-organizing, collective dynamics on the scale of an entire society. Functioning as a "super organism"[1], a honey bee colony can contain thousands of individuals whose intricate behavior results from a shared genetic background and sophisticated social signals conveyed through multiple communication channels[2]. The effect of these dynamics is to cooperatively divide and organize the effort necessary to maintain a well-functioning collective in response to external and internal environmental change, thus enabling the colony to grow and reproduce. A longstanding fascination with such behavior has driven substantial previous work (see e.g., ref. [3]), which includes examinations of collective behavior[4,5] also in combination with high-throughput sampling technologies such as gene expression sequencing[6–8]. However, a full quantitative understanding of the colony behavior requires measuring the collective dynamics at single-organism resolution as well as the spatiotemporal patterns of colony resources such as food and brood. Both of these challenges are now accessible through advances in machine vision.

A honey bee colony contains a high density of visually similar members, rapidly moving and occluding on the uneven and changing honeycomb surface, and whose numbers change in time. These factors present substantial difficulties for automated image analysis[9–11] for which a common solution is to apply physical tags to some[12] or all[13–17] of the colony members. Barcoded tags allow for the distinct marking of a sufficiently large number of individuals to track a naturally sized colony and have been exploited to unravel important aspects of bee communication[12,18] and information spread[14,19]. This information together with recognition of individual bee behaviors bring insights into the quantitative understanding of a bee colony. While a wealth of information can be extracted with the use of tags, the burden of manually tagging hundreds or thousands of small insects, without harm or inhibition to their motion, does carry some limitations. For example, marking newly hatched bees requires either opening the hive or introducing marked newborns without letting any hatch inside, both of which disrupt the colony[13,18]. Moreover, the recognition of markers is hampered by visual occlusions of parts of the tag and image blur. In the dense environment of a hive, even when confined to a 2D surface, partial occlusions are common. In particular, behaviors such as crawling inside of a honeycomb cell, or walking upside down on the glass of the observation hive, can obscure the marker, thus limiting behavioral repertoires captured with a marker-based approach. The difficulty and workload of manual tagging also hinders the analysis of multiple colonies for extended timespans, as is routinely accomplished in behavioral studies of other organisms.

Recent breakthroughs in image analysis using CNNs, including fast and accurate single and multiple object detection[20–22], posture quantification[23–26] and image and video appearance representation[27–29], offer new inspiration and opportunities for extracting information directly from video data in dense-animal contexts. However, most existing solutions and benchmark datasets for multi-object tracking as well as for posture and activity recognition, are dedicated to human behavior and crowds[30–34]. In biological image data CNNs have been broadly exploited for cell or particle segmentation[35–38]. Versatile, supervised CNN-based tools have also been proposed for animal posture quantification[25,26,39–41] and have been successfully applied to the study of insect behavior[42–44]. While facilitating important tasks in bioimage interpretation, these solutions are limited to behavior of few individuals and do not resolve challenges within the task of dense object detection and tracking in a bee colony[45,46].

Importantly, previous work has shown that seemingly identical organisms do carry distinct visual features, also termed 'pixel personality', which can be quantified and leveraged for markerless tracking[47–51]. The rise of deep learning has enabled new ways of extracting such visual signatures from images. However, using CNNs for the quantification of pixel identities requires a set of instances of each object that can serve as a training set. For example, in tracking up to 100 fish such a set could be constructed if a segment in a video existed where no fish collisions were observed during 3000 video frames[50]. In the much denser environment of ~1000 individuals inside of a beehive, such collision-free segments are rare or nonexistent. A previously proposed solution[49] extracted pixel identities of bees based on a smaller number of instances, re-learning these identities after each trajectory matching step via a classification CNN architecture[52]. Starting with a set of short trajectories the classification network is trained to assign a different label to bees belonging to the separate trajectories. After a set of training iterations, the detections in the following video frame are labeled by the network. Detections, which are assigned a label with high confidence are added to the respective trajectory. The network is then retrained on the expanded train set. Unfortunately, this solution offered only limited accuracy and came with a high computational cost, limiting its practical use.

Here we expand the use of CNNs to capture, at single-organism resolution, the colony-wide composition and behavior of the honey bee *Apis mellifera*. Our approach applies to images and video recordings of unmarked colonies and enables broad quantitative study without the burden of manual marking. We demonstrate our solution through the analysis of five long-term timelapse recordings, up to four months in duration (segmentation and sociometry), as well as of five short-term videos recorded at high frame rate for 5 min (tracking and motion behavior). The data were collected at multiple locations on the campus of OIST Graduate University (Okinawa, Japan) with varying imaging arrangements. We infer the position, orientation and within-cell state of each bee together with the location of brood cells (Fig. 1a). Using these detection results we devised a neural network and an efficient training method for quantifying visual features capturing similarity among bee instances (Fig. 1b). We use this similarity to stitch bee detections into trajectories across difficult occlusion and touching events. Our solution resolves confounding trajectories locally in time and space, though the reidentification of a bee after leaving the hive is not yet possible. We demonstrate our approach with long-term sociometric monitoring (detection) and with colony-wide exploration of individual behaviors (tracking). Along with this manuscript and associated code, our contribution includes the labeled image data of thousands of trajectories of bees in dense configurations, a unique resource, which offers opportunities for both machine vision and biological research.

## Results

**Long-term colony sociometry**. To capture the long-term sociometric dynamics[53–56] of bee colonies, we devised a segmentation-based method for the detection of bees and brood in dense configurations within a 2D hive[21,49]. We reported the honey bee (but not brood) detection in an earlier conference proceedings[21] and we review that approach here for clarity. In brief, our solution is based on a modified segmentation CNN architecture[20], which exploits the temporal dimension of video data to improve accuracy (Fig. 2a–c). During training and inference this network uses information from the preceding video frame, allowing us to reduce the size of the network by 94% compared to the original

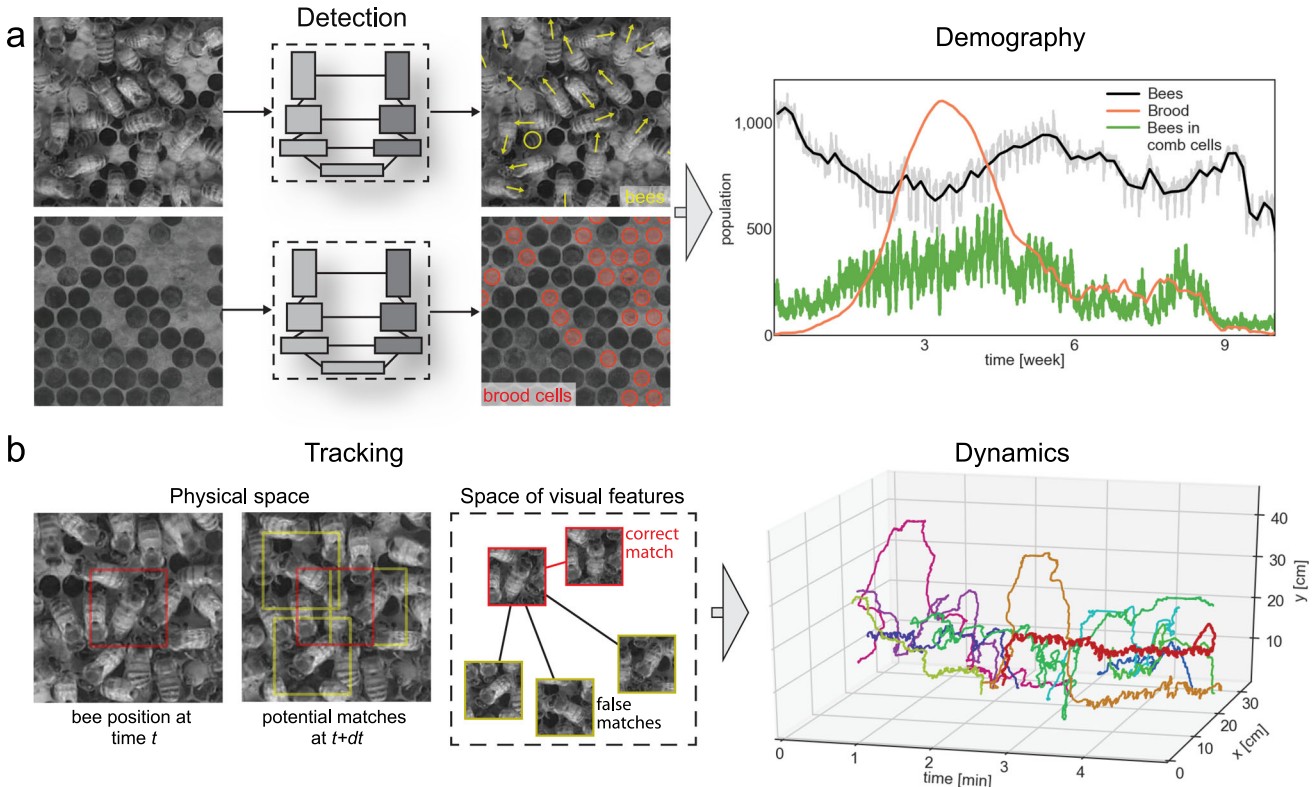

**Fig. 1 Schematic of the detection and tracking methods. a** A segmentation architecture was used and adapted independently for two tasks: the detection of bees and of brood cells in the dense environment of a bee colony. We use this architecture to infer bee and brood positions (yellow and red marks, respectively), bee posture type—bee inside of a comb cell (yellow round symbols, denoted later as cell-bees) and fully visible bee (yellow arrows, denoted later as full-bees)—and the orientation angle of the fully visible bees (angle of the yellow arrows). Bee and brood locations, as well as within-cell state, allow for detailed sociometric analyses over long timespans and across multiple colonies (right). **b** CNN-derived embeddings of bee images are used to link detections across video frames into trajectories. The network is trained to maximize the Euclidean distance between detections belonging to different trajectories and minimize this distance between detections of the same bee within one trajectory, which is symbolically illustrated in the middle panel. This visual similarity metric allows for accurate construction of trajectories of unmarked bees and analysis of dynamic aspects of bee colonies. (right) For illustration we show ten tracked trajectories, likely corresponding to forager bees. The red trajectory belongs to a bee that entered the hive around minute 2 of the recording and performed rapid back-and-forth dancing motion for the next 2 min.

architecture while still achieving high detection accuracy. Within one network we infer both the segmentation maps through pixel classification as well as the orientation angle of the segmented objects through regression. Each pixel is classified into one of the three categories: 'background', 'fully visible bee' (denoted full-bees throughout the text), and 'bee inside comb cell' (denoted cell-bees throughout the text) if only the bee abdomen is visible. The accuracy of the detection method was assessed on a large set of manually labeled images[21]. We additionally inspected the performance of our detection method in three frames from recordings L5 and S5, after fine-tuning of the network by retraining it on a set of up to five frames from recordings L1–L4 and S1–S4. In frames from the beginning, middle, and end parts of recordings L5 and S5, recordings that were not used in the retraining, we estimated detection true positive rate (TPR) at ~0.99, false positive rate (FPR) at ~0.03 and false negative rate (FNR) at ~0.01 (Supplementary Fig. 1, Supplementary Table 1). These results confirm the capacity of our detection algorithm to generalize across recordings registered in different hives and imaging setups and also provide a strong foundation for further analysis and tracking method development.

We devised a similar segmentation-based approach to locate capped brood cells by exploiting background images automatically generated for hive images over timespans of 12 h (Supplementary Movie M1). This solution did not include the recurrent element in the network design and relied on the original segmentation

architecture[20] (Fig. 2d–f). To mark brood cells we used round-shaped segmentation markers (Fig. 2f). Our method was trained on three initial images from recordings L1–L4 and generalized to the rest of these recordings as well as to the recording L5 (Supplementary Fig. 2, Supplementary Movie M2) with performance of TPR ~ 0.99, FPR ~ 0.01, FNR < 0.01 evaluated in the frames shown in Supplementary Fig. 2. Previous work[57] has proposed a similar approach to the detection and recognition of all cell types (brood, honey, nectar, pollen, larva, egg), however, in a standardized imaging setup and based on empty frames that were removed from the hive. While our imaging arrangement and the noise introduced by the background extraction algorithm currently do not provide the detail needed to detect eggs or pollen, our approach does provide access to continuous measurements of the brood counts in a living colony.

We deployed both detection methods on a set of long-term video recordings (Supplementary Table 2) and extracted quantitative measures of demographic changes in bee colonies over periods ranging from two weeks to four months. In all long-term recordings (L1–L5) we found that the total number of visible bees (full-bees and cell-bees together) as well as cell-bees undergo repetitive fluctuations (Fig. 3a, b, Supplementary Figs. 3, 4). The period of these fluctuations is close to 24 h based on spectral analysis (Fig. 3a, b, Supplementary Fig. 5). Over longer times, we observed an example of high visible bee counts occurring after a peak in brood number (Fig. 3c). We also found a strong negative

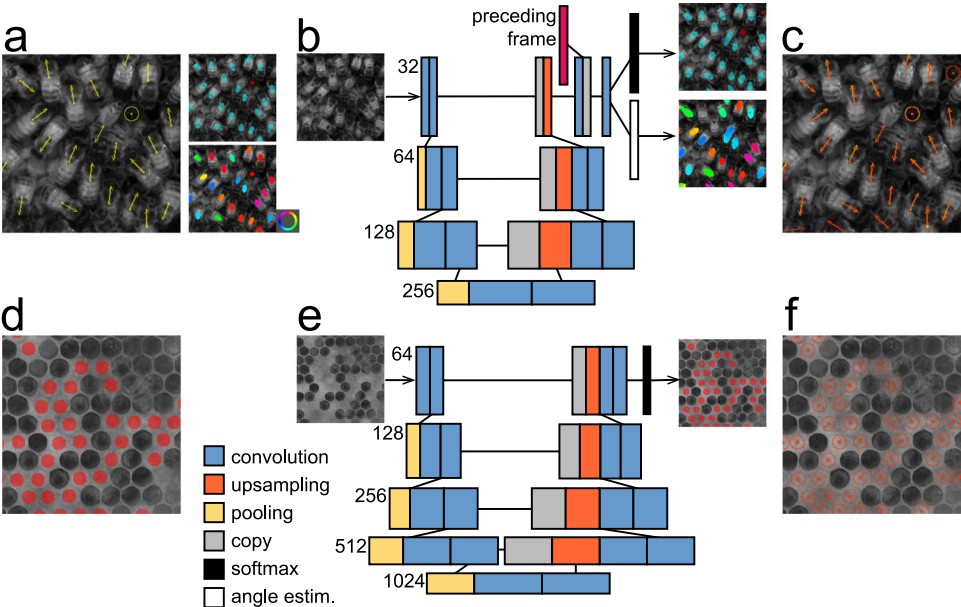

**Fig. 2 Dense object detection. a** Our manual annotation consists of the central points of each body, object class—full-bees (yellow arrows) or cell-bees (yellow round symbol)—and the body orientation of full-bees (arrows). Segmentation maps are created with foreground pixels denoting central parts of bee bodies. The upper segmentation map indicates class information, the bottom segmentation map indicates the orientation angle (color wheel). **b** We modified the U-Net architecture[20] by including a recurrent component which exploits temporal information from the preceding video frame in the penultimate network layer. The recurrent component allowed us to reduce the number of parameters by ~94% from the original U-Net (shown in **e**) without compromising accuracy. We added two loss functions to the network, one for class and one for orientation angle estimation, and their example output is shown to the right. In panels **b** and **e** the information flows from the left (image) to the right by passing through the lower layers of the U-Net to produce the segmentation maps on the right (segmentation map). The numbers in **b** and **e** indicate the number of filters of the convolutional layers for each level of the U-Net. **c** Position, class, and orientation information is inferred from the network output. Red markers indicate inferred detections, labels are in yellow. **d** Segmentation maps created for the training of brood cell detection. Similar to body detections, the foreground pixels cover only the central parts of the cells, allowing for object counting and localization. **e** The original U-Net architecture is used for brood cell detection. **f** Inferred brood cell positions.

correlation between numbers of visible bees and brood counts measured in 12 h windows (Fig. 3d, Pearson mean $R^2 < -0.84$, $p < 0.0001$ over the first three weeks), which could be an indication of homeostatic control of colony size[58–60].

We next investigated daily changes in the number of full-bees and cell-bees as potentially reflecting the daily foraging, brood nursing, and other activities in each colony. We found high numbers of cell-bees predominantly at night (Fig. 3e) which may represent sleeping bees[61]. This regularity was not apparent in the visible bee counts, where the oscillations varied, perhaps due to external factors (Supplementary Fig. 6). We thus inspected the weather conditions during the time of recordings and, while no extreme events were observed (Supplementary Fig. 7), two colonies (L3 and L4) were filmed during particularly high temperatures exceeding 26 °C at night, which can encourage bees to stay at the hive entrance[62] where they would not be detected. The altered daily behavior could also be ascribed to factors we do not track in this study, such as toxins in the environment[63].

While we found insufficient evidence linking the phase of the visible bee number change to the weather conditions, we did find that full-bee numbers shift in relationship to brood presence. During periods when brood numbers are high, the number of visible bees tends to be in phase with the cell-bee number with high numbers at night. Such a pattern potentially corresponds to the foraging activity of the bees performed uniquely during the day[3] and it is more prominent in hives with high brood numbers, which might reflect the increased food demand of growing colonies (Supplementary Fig. 8).

Similarly, the cell-bee number may be related to brood presence. In three of the five long-term recordings the proportion of cell-bees was positively correlated with the brood count (L3–L5,

Pearson $R^2 > 0.4$, $p < 0.05$). We also found that the cell-bees spatially significantly closer to the brood cells than other bees (by 7 to 14 mm on average, Wilcoxon test $p < 0.0001$ in L1–L4) and this distance decreases during the night and increases during the day (Supplementary Fig. 9). No storage cells could be observed in the vicinity of brood in these colonies.

In two of the long-term recordings—L3 and L5—the colonies experienced a systematic decrease in the number of individuals. In L3 we noticed a moth infestation in the wax towards the end of the recording (Supplementary Fig. 10) which might have weakened the colony. In recording L5 each subsequent increase of the brood and colony size was lower than the previous one (Fig. 3c) ultimately leading to the gradual depletion of the colony. Despite an initial rapid increase in brood numbers, another wave of brood of such size did not occur and the colony never recovered its initial size. In hives L3 and L5 the proportion of cell-bees was noticeably lower (Fig. 3f). The cell-bees in those hives were additionally located much farther away from the brood cells than in the healthy colonies (Supplementary Fig. 9).

While the specific reasons for the decline of these colonies are unclear, we show quantitative measures indicative of behavioral changes in unhealthy bee colonies. Indeed, the colony collapse phenomenon currently threatening bee populations worldwide have been ascribed to a range of factors[64,65]. However, despite numerous studies and theoretical models[66–68], no systematic, quantitative, and data-driven assessment has yet been proposed. Unraveling the exact reasons for and mechanism of honey bee colony collapse would require multiple comparative observations across a range of conditions. Our approach enables the study of this phenomenon in a time-resolved, comprehensive, and quantitative manner.

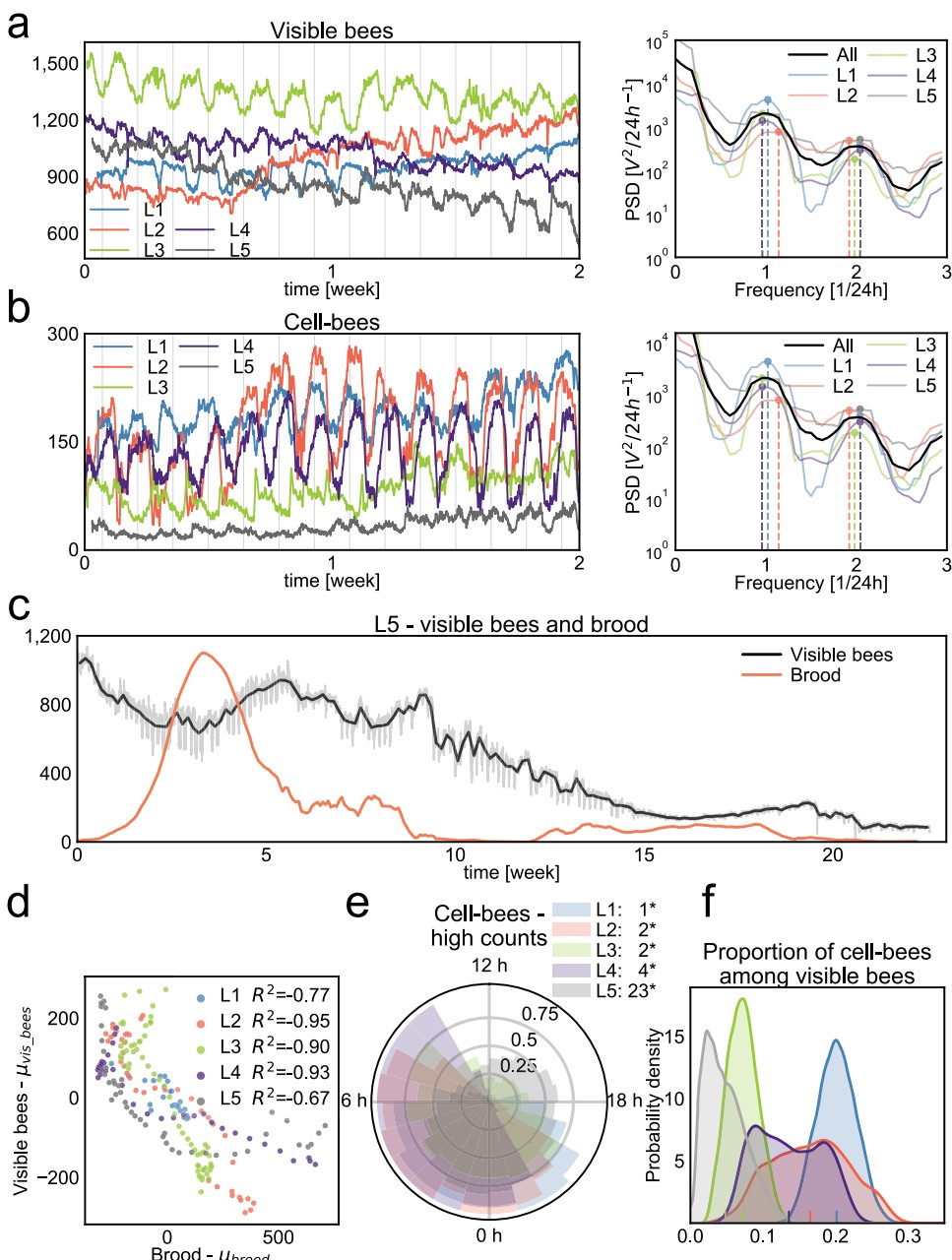

**Fig. 3 Colony sociometry. a** Daily fluctuations of the visible bees (full-bees + cell-bees) in hives L1–L5, where vertical lines denote midnight. The period of the fluctuations is approximately 24 h in all hives, which is indicated by the power spectral density (PSD, right). Dashed lines in the PSD denote two highest values in the density plot of each hive. **b** Analogous to **a**, daily fluctuations in the numbers of cell-bees and the respective power spectral density. **c** Visible bee numbers (full-bees + cell-bees, gray and black lines) and brood numbers (orange line) in bee colony L5. Colony L5 was imaged for over 4 months and exhibited a decline in bee numbers and ultimately colony collapse. After an initial rise in brood number, another increase of such amplitude did not occur, and the colony steadily declined from week 10. The black line indicates the daily average number of visible bees while the exact count in each image is shown in gray. **d** Bee and brood numbers are anticorrelated. Total bee number averaged over 12-h windows is plotted against the brood counts in the same time window in the five colonies for the period of up to 3 weeks. For each time series, the mean of the series was subtracted from the plotted values. **e** The nightly presence of cell-bees. We divide each 24 h of a recording into 24 1-h bins and count how often the number of cell-bees is above the daily median number. Numbers in the legend indicate mean time of the day when the high counts are observed. Asterisk ('*') symbol marks hives where the observed high counts are not uniformly distributed based on $p < 0.0001$ in the Rayleight test of uniformity[97]. High numbers of cell-bees occur predominantly in the evening, between 21 h and 6 h. **f** Distribution of the proportion of cell-bees relative to the total number of visible bees in hives L1–L5. Hive L3 and L5 show markedly lower proportion of cell-bees, which may be related to colony declines during our recording. Source data are provided as a Source Data file.

**Colony-wide tracking at single-organism resolution.** Our high detection accuracy provides a strong foundation for reliably following objects in time. Rather than focus on distinct markers of each individual[47–49], we apply a rich representation of visual features to explicitly resolve confounding trajectories locally in space and time. Quantitative visual features are extracted from a CNN[52] trained using a triplet loss objective function[69,70]—a function which enables the expression of similarity among entities via their vector embeddings. During training, triplets of images are fed into the network and we search for a solution

where the Euclidian distance between vector embeddings of two images of the same object at different time points is small as possible while the distance between vector embeddings of two different objects is large (Fig. 1b). The major hurdle in training such a network for bee tracking is the massive space of all possible triplets containing two instances pertaining to the same trajectory and a third instance pertaining to any other timepoint of any other trajectory. Systematically training on a complete set of all possible triplets in our data is not feasible in a realistic time. To address this problem and enable network loss function convergence in a reasonable time, we devised a dedicated training scheme. Briefly, image triplets were sampled from training set trajectories and included only bee detections neighboring in time and space, which are precisely the sources of potential identity swaps. Additionally, the embeddings of all triplets used during training were simultaneously tested for correctness and the incorrect ones—where the distance criteria between positive and negative matches were not met—were used again in the network training (Supplementary Fig. 11). For detailed training method description, see the Methods section.

We trained the network on a large, incrementally constructed set of validated bee trajectories. We used four 5-min-long recordings (S1–S4) captured in different locations on the campus with cameras of varying pixel resolutions (Supplementary Table 2). While the hives were all the same spatial size, the colonies varied in number of detected organisms, ranging from $N = 805$ to $N = 1316$, and in their dynamics (Figs. 4a, 5a). All videos were recorded during daytime, under good weather conditions, and within the foraging season of Okinawa.

We first applied a "pixel personality" approach[49] on recordings S1–S4. The trajectories resulting from this approach were manually validated, and the correct trajectories formed the'initial dataset'. This dataset was used for training the network with the

triplet loss described above. Visual feature embeddings extracted from this CNN incorporate similarity among individuals by combining orientation, posture and background information (Fig. 4b, Supplementary Movies M3–M7, Supplementary Figs. 12, 13) and hence allow the linking of bee detections across video frames in presence of temporary object occlusions. We implemented a detection matching solution exploiting both position and visual feature embeddings, and matchings are done in a greedy manner on a sorted list of all pairs satisfying predefined time and space proximity criteria (see Methods).

The trajectories constructed with the use of embeddings from the triplet loss-trained CNNs were constructed in videos S1–S4 and manually validated. Compared to a pixel identity approach[49], our solution offered an important increase in accuracy at a low computational cost (Fig. 4c). Visual feature-based matching resulted in a higher number of correct trajectories over the pixel personality approach as well as over a position-only solution (Fig. 4c, Supplementary Figs. 14, 15). Most trajectories belonging to these recordings were part of the training set, nevertheless this result suggests the method's capacity to generalize to other, trajectories not included in the train set within the same colonies. As an estimate of computational cost, the detection and trajectory matching for one 5-min-long recording completed in ~1 h on a 4 GPU, 36 core CPU machine.

We next used the entire set of trajectories validated as correct in videos S1–S4 to form the 'final dataset'. Embeddings derived from the network trained on the expanded training set delivered an increase in tracking performance in videos S1–S4 (Fig. 4c), emphasizing the role of the training set size in producing more precise and distinct embeddings. We tested a range of data augmentation scenarios, background masking (Supplementary Fig. 16) and we additionally used orientation angle in the matching procedure, none of which improved tracking accuracy

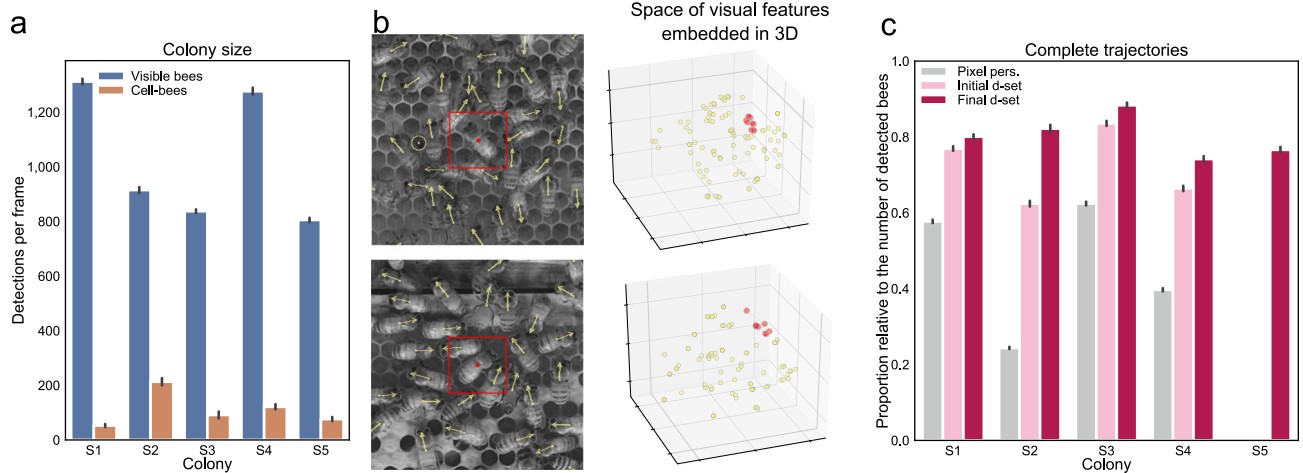

**Fig. 4 Leveraging visual features for enhanced detection matching across frames. a** Recordings from five different beehives were used for tracking method development and testing. The hives contained varying total numbers of detected bees and of bees in comb cells (cell-bees). The colored bars show the mean number of detections across all frames ($N = 3600$), the error bars show their standard deviation. **b** CNN-derived vector embeddings are used to better match bee detections across video frames. Even though bees appear identical to a human observer, the embeddings belonging to one trajectory (red) are distant from embeddings of all other bee images neighboring in time and space, thus aiding the correct stitching of individual detections. The embeddings encode similarity among images of detections. Originally 64-dimensional, an example projection of the embeddings in 3D is shown in the panels on the right obtained with the use of t-SNE[98]. Red dots represent embeddings in 10 consecutive video frames of the bees marked by red squares in the left panels. Embeddings of other bee detections in these images that occurred over consecutive three video frames are marked by yellow dots in the panel on the right. **c** The accuracy of the tracking method obtained for test recording S5 matched the accuracy reached in recordings S1–S4 used for network training and method design. We show the proportion of extracted trajectories relative to the average detected number of bees in hives S1–S5. The proportion of extracted trajectories is shown for an earlier approach ('pixel personality', gray bars), the current method but trained only the initial dataset (pink bars) and for the current method but trained on the final dataset (red bars). The colored bars show the proportion relative to the mean number of detections in all frames, the error bars denote the standard deviation in proportions relative to the detection number across all frames ($N = 3,600$). Source data are provided as a Source Data file.

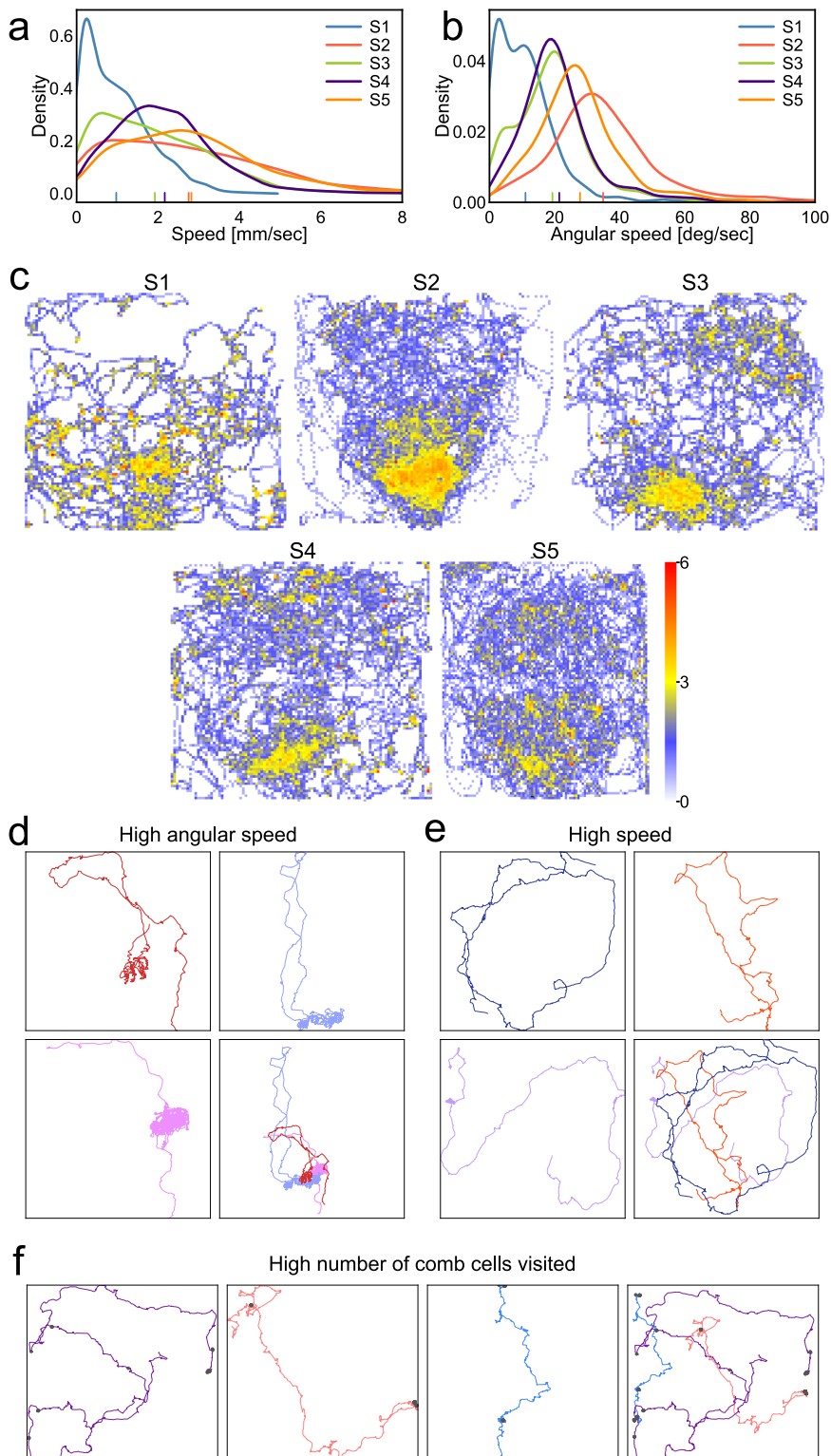

(Supplementary Fig. 17). We attribute this to the fact that data augmentation importantly increases data complexity hindering network convergence. Poorer matching accuracy on cropped images suggest that the background, including other neighboring bees, and body parts such as legs and wings play an important role in correct matching. We additionally used the collected dataset of validated trajectories to test the limits of our method to make correct matchings if the video frame rate is reduced (Supplementary Figs. 18, 19).

To test the capacity of our method to generalize to other recordings, we first used the recording S5. No images of this colony were used in the training of the detection or tracking method. Recording S5 is additionally characterized by vibrations due to a neighboring construction site and flickering of the lights, creating particularly challenging tracking conditions. We found that 77% of detected bees were correctly tracked (Fig. 4c), a result comparable to the recordings that were part of the training set (Supplementary Table 3). To further corroborate the generalization of tracking

**Fig. 5 Colony dynamics at single-organism resolution. a** The distribution of individual mean speed computed from trajectories in hives S1–S5. Large differences across hives include a significant proportion of immobile bees in S1 and increased number of fast-moving bees in S2 and S5. The vertical lines (also in panel B) indicate the mean value of each distribution. **b** The distribution of individual mean angular speed of trajectories in hives S1–S5. Low angular speeds are seen primarily in S1 while the largest proportion of trajectories with high angular speed is present in S2. Trajectories in the tail of these distributions are excellent candidates for forager bees performing or following a waggle dance. **c** The spatial distribution of trajectories characterized by large linear and angular motion. We show 100 trajectories from each hive which have large linear and angular speed. The heatmap illustrates counts of these bees' appearances in each point of the hive. The entrance is located at the bottom of each hive. Across all hives these trajectories are located near the entrance and this localization may reflect a forager recruitment site. **d** Example trajectories of individuals with large linear and angular motion. Three trajectories are plotted individually and combined in the bottom-right panel. The densely overlapping parts of these trajectories indicate the location of the waggle dance performed by these individuals. **e** Example trajectories showing large linear but low angular motion. Such individuals tend to move rapidly over large portions of the hive. **f** Example trajectories of bees exhibiting a large number of comb-cell visits. Source data are provided as a Source Data file.

performance, we performed a cross-validation test in which each of the recordings was taken out of the train set, then the network training and calculation of embeddings was performed using the other four recordings. Such cross-validations resulted in comparable results of 70–86% of correctly tracked bees across the recordings (Supplementary Fig. 20). Such high accuracy provides a strong foundation for the study of colony dynamics and we expect improvements as more validated trajectories are incorporated in the training set. The entire set of correct trajectories assembled during development of this method includes 4642 trajectories from the short-term recordings S1–S5 and is provided with this manuscript as an important resource for further study and method development.

**Quantitative analysis of colony behavior**. The large set of trajectories generated through our detection and tracking approach covers an extensive proportion of the recorded colonies (Supplementary Movies M8–M12) enabling a broad, comparative study of honey bee colony dynamics. While our emphasis here is on the techniques of detection and tracking, we can already offer a variety of quantitative observations.

We first compared aggregate dynamics using the distributions of individuals' speed, angular speed, diffusion coefficient, as well as motion span defined as the diagonal of the minimal rectangle fitting the trajectory and overall path distance (Fig. 5a, b, Supplementary Figs. 21, 22). Colonies S1 and S3, despite differing in the number of individuals (1316 and 839, respectively), are characterized by a large proportion of motionless individuals and individuals moving over only limited areas of the hive (8% of the colony diagonal on average in S1). Whereas colonies S2 and S5 showed a high proportion of fast-moving bees and bees traversing larger portions of the surface of the hive (~24% of the hive diagonal on average). Additionally, colonies S2 and S5 contained a higher proportion of individuals moving at high angular speed (Fig. 5b) which could be indicative of the foragers performing waggle dance[2].

We next inspected the spatial localization of the individuals showing the highest values in each motion attribute (speed, angular speed, area of motion, etc.). Most prominently, bees showing the highest angular speed tend to be located at the bottom part of the hive in proximity of its entrance (Fig. 5c). These bees are ~2.5 times more probable to be located within 10 cm radius from the hive entrance ($p < 0.001$, permutations) and bees from hive S2 and S5 are significantly overrepresented among the 100 fastest trajectories ($p < 0.01$, permutations). The colocalization of these individuals, most noticeable in S3, is present across all colonies, suggesting that relatively high angular speed is a characteristic of foragers performing and following the waggle dance at the hive entrance[71].

To corroborate this hypothesis, we manually inspected trajectories showing the highest mean values of single motion attributes as well as their combinations. Among manually inspected 100

trajectories with highest angular and translational motion we found that dancers and dancers' followers are significantly linked to higher motion velocities ($N = 10$, $p \sim 0.003$ and $N = 27$, $p \sim 0.025$, respectively, Wilcoxon rank sum test corrected with permutations, Fig. 5d). The high angular speed of these individuals is displayed during the fast-looping dance motion of the foragers indicating to other colony members locations of food sources (Supplementary Movies M13–M15). In contrast, individuals with high translational motion but low angular velocity tend to visit large portions of the hive without performing any recognizable action (Fig. 5e), a potential sign of patrolling behavior[72]. We additionally quantified the number of times a bee in each trajectory visits a comb cell and identified bees that clean or search through comb cells sometimes across long distances (Fig. 5f, Supplementary Movies M16–M18). These examples demonstrate that the collected metrics group together similar and potentially meaningful individual bee behaviors and could facilitate their fully automated detection in the future.

## Discussion

Recent machine vision advances in the precise posture tracking of individual animals[25,26,39] as well as of the positions of highly-similar organisms in groups[50] are enabling quantitative studies of behavior[73,74]. In collective behavior specifically, the use of CNNs for the pixel-based identification of individual organisms has significantly advanced markerless, long-time tracking in 2D, from more modest assemblies (~10 individuals)[48] to larger groups (~100 individuals)[50]. However, while network-learned identities can resolve confounding visual occlusions and overlaps, a principal challenge of individual-resolution group tracking, there must also be enough isolated instances to train the identification network. These conditions are rare or nonexistent in the dense and cluttered setting of a honey bee colony, where visual occlusions and overlaps are perpetual. Here we have described a detection-to-dynamics solution, which expands tracking to large and dense collective systems.

Our markerless detection and tracking techniques offer possibilities for the quantitative study of honey bee colonies on the collective scale at single-organism resolution and are complementary to existing approaches[10,16–18] in particular by tracking organisms, which are difficult or impossible to tag. Our bee detection solution exploits temporal information for improved accuracy. Brood detection could be improved in a similar manner by including temporal information and the information on the expected lifetime of capped brood. Both the brood and bee numbers change over time (Fig. 3), a challenge for manual marking, and (with improved lighting of the observation hive) our detection technique can be readily extended to include capped and uncapped honey cells and cells as well as egg and larvae that are not currently detected. The dynamics and spatial arrangement of these variables provide quantitative data for sociometric analysis[53–56] and will be particularly interesting in

the context of a collapsing colony. Colony-wide, high-resolution tracking augments larger-scale measures such as weight[75] and can be combined with additional hive sensors for a novel surveillance system[76]. The automatic nature of our approach also facilitates the imaging of multiple hives[77], an important consideration due to colony-to-colony variability.

Our trajectory construction method currently spans over 5 min, an interval chosen to enable extensive manual validation of the results (Fig. 4). We expect that this interval can be increased with more training data (trajectories from more colonies), advanced validation techniques (such as using markers invisible in the infrared), as well as with improved matching procedures. In particular, learning motion patterns via a recurrent neural network[78] can improve matching accuracy over the heuristic approach proposed here. Combining motion and appearance learning, while still underexplored[79], has potential to result in a single and powerful tracking solution. Expanding the beehive to include two sides will allow for monitoring of larger, healthier colonies, although care will be required to track individuals when they change sides. An important outcome of this work is to provide first techniques together with rich trajectory data for further improvement as machine vision methods accelerate.

A wealth of behavioral information is already accessible within our current tracking window. While not a target of our study, we could readily detect bees performing the waggle dance based on their motion (Fig. 5b). Other possibilities include detection of sleep[80], trophallaxis[81], fanning and scenting[82], for which additional appearance cues can be used, such as the extension of the proboscis and wings. New metrics based on detected behavior, such as amount of nursing, sleep or foraging, or patterns in communication signals, could become instrumental for assessing of colony health. Additionally, the quantitation of behavior and aggregate colony dynamics over short time scales could be used in combination with the long-term sociometric observation of the same colonies as well as with long trajectories of selected individuals obtained via tagging. The accuracy of CNNs in detection of visual detail together with the large numbers of trajectories of unmarked bees offered by our methods present vast opportunities for behavior analysis of bees and open avenues for more quantitative approaches to modeling colony dynamics[83–87]. We also see no obvious obstacles in generalizing our approach to other dense insect collectives such as ants.

Across the organizational scales of living systems, from molecular, through neural[88], to animal groups and societies[16,89], including humans[90], our ability to understand emergent collective behavior has been significantly enhanced by modern precision measurements and analysis across large portions of the ensembles. With the advances reported here, we expect accelerated progress in our understanding of the behavior of honey bee colonies and other crowded systems.

## Methods

**Imaging setup**. We situated observation hives in two distinct locations: location 1 on the rooftop of an OIST building (~4th-floor elevation), and location 2 in a ground-level shipping container, which was also surrounded by greenery. Both locations were equipped with infrared LEDs and a heating system, which maintained a constant room temperature of 31 °C. The LED system was composed of four panels of $220 \times 220$ mm size illuminating the hive from four different angles (bottom-left, bottom-right, top-left, and top-right) at a distance of ~1 m and with the wave length of 850 nm. Each panel was equipped with 14 stripes (6 LEDs/strip) for a total of 84 LEDs and 13.4 W per board. The observation hives were $47 \times 47$ cm in size, fitting two honeycomb frames placed one above the other. The back side of the comb was fixed to a wooden surface constraining the bees to only one side of the frames. In location 1, images were obtained with a $5120 \times 5120$ pixel Vieworks Industrial Camera VC-25MX-M72D0-DIN-FM, at 30 FPS. In location 2, images were obtained with two lower resolution cameras Panasonic Lumix GH5 4 K and Blackmagic Design Production Camera 4 K at 30 FPS with a typical 4k resolution of $3840 \times 2160$ or $2560 \times 2560$ pixel for the long timespan recordings. Images of recordings from location 1 (S1, S2) were spatially downsampled by a

factor of two resulting in a similar pixel-per-bee resolution for the recordings from both locations. All cameras were modified for infrared imaging by removing the infrared filter.

**Detection dataset**. For the development of the detection method we generated two recordings in location 1 (D1, D2). Two sequences of 360 images from each of these recordings, were used for training and testing. We devised a custom labelling interface (https://github.com/oist/DenseObjectAnnotation) for manual annotation of bee locations and orientations which was used to label images in D1 and D2. A subset of four frames—two from each recording with 2034 bees was additionally labeled 10-times by independent labelers to obtain an estimate of human error in position and orientation labeling. This error was calculated as the standard deviation of distance of each of the 10 labels to the reference label obtained in the main labeling task and additionally error checked and corrected by us.

The annotated dataset contained 375,698 labeled bees. Every bee was assigned $(x, y, b, \alpha)$ denoting the coordinates of its central point against the top-left corner of the image, type of the label ($b = 1$ for full-bees and $b = 2$ for cell-bees), and the body rotation angle $\alpha$ against the vertical pointing upwards and calculated clockwise ($\alpha = 0$ if $b = 2$). We generated pixel regions centered over the central point of each bee (Fig. 2a). For full-bees the regions were ellipse-shaped, for cell-bees circular, both with size of a third of the bee dimensions in the image. Such regions cover central parts of each bee and are nonadjacent to regions covering neighboring bees in the image. By choosing a small size of the foreground pixel blobs we minimized the possibility of overlap of blobs marking individuals in dense configurations.

These foreground regions were assigned values of class b in the classification segmentation maps. Background pixels were assigned value 0. For learning of the orientation angles, each foreground pixel, instead of class label, was assigned the value of the bee rotation angle and the background pixels were labeled as −1. To compensate for the class imbalance between foreground bee regions and the non-bee background, we generated weights used for balancing the loss function at every pixel. For every bee region a 2D Gaussian of the same shape was generated, centered over the bee central point, and scaled by proportion in the training set of the background pixels to the number of bee-region pixels.

For training and testing the images were organized in 60 sequences of 360 images of $512 \times 512$ pixel size. In this time-resolved data the first 324 images of each sequence were used for training and the remaining 36 for testing.

**Detection network and training**. We used the U-Net segmentation network[20] and expanded its functionality to take advantage of regularities in the image time series. In each pass of the network training or prediction the penultimate layer was kept as a prior for the next pass of the network (Fig. 2b). In the following pass the next image in the sequence was used as input and the penultimate layer was concatenated with the prior representation before calculating network output.

We used two loss functions in the last network layers. The first loss function, the 3-class softmax function, allowed for performing foreground-background segmentation. The second loss function was defined as $L = \sin\left(\frac{\hat{\alpha} - \alpha}{2}\right)^2$ where $\hat{\alpha}$ and $\alpha$ are the predicted and labeled orientation angle, respectively.

In the network output, each contiguous foreground region was interpreted as a mark for an individual bee. Foreground patches smaller than 10 and larger than 1000 pixels were discarded as potentially wrong. The centroid location was calculated as the mid-point of all x and y coordinates of points in each region. Region class was assigned as the class identity of the majority of pixels within given region. We calculated the main body axis of full-bee regions as the angle of the first principal component of the points in each region. The orientation angle was predicted only for the full-bee class and was calculated as the principal axis angle oriented based on the predicted angle valued of the pixels within a given foreground region. Due to the higher prediction errors in the image margins, where objects are not fully visible, during inference, we used windows, which overlapped by a margin of 50 pixels. Any objects in the image margin were discarded from the results.

The resulting segmentation maps allowed us to find individual locations in an independent test set of D1 and D2 recordings with an error of <10% of a bee body width, detection TPR ~ 0.96, FPR ~ 0.14 and orientation angle error of ~9.7° closely matching error of human labelers (Supplementary Table 1).

**Recordings**. All image data were collected from spring to fall in favorable weather conditions when foraging activity was observable. For each recording we typically selected from the hives in our apiary two honeycomb frames with even surface, one containing brood and one containing food stores. We removed comb cells and other content on one side of each frame and left them for another day inside of the apiary hive to allow the bees to clean the damaged surface. Next, after ensuring that the queen is located on one side of each frame, we transferred them into the observation beehive and fixated their empty sides to the back surface of the hive. Before recording, we allowed each colony to adjust for 2 weeks after the transfer from the apiary to the observation hive.

We performed long-term timelapse imaging of five colonies (L1–L5) for a timespan between 2 weeks to 4 months (Supplementary Table 2) with sampling frequency of once per minute (colony L5) and once every 2 min (colonies L1–L4).

We additionally imaged five colonies (S1–S5) with a sampling frequency of 30 FPS. For the tracking analysis we used five-minutes segments, which we downsampled to 10 FPS.

We trained the bee detection network as described in the section above on the data and labels of D1 and D2 scaled down by a factor of 2. This rescaling was necessary so that the body sizes of bees in pixels in D1–D2 matched their size in the recordings L1–L5, S1–S5. We next generated predictions on several initial frames of recordings L1–4 and S1–4.

To compensate for potential noise introduced by image rescaling and to incorporate images of empty parts of the hives in the training set we used the labelling interface (https://github.com/oist/DenseObjectAnnotation) to correct the bee position and orientation angle predictions in five initial frames of each recording. We then retrained the detection model for 10 training iterations on the additional set of labels. In this way, with a relatively small amount of manual labeling, we could adapt our method to the new recordings (Supplementary Fig. 1, Supplementary Table 1).

**Brood detection**. We devised a similar segmentation-based method for detecting capped brood cells in a colony. We used a background extraction method[91,92] applied to a range of images spanning 12 h resulting in one for each consecutive 12 h of a timelapse recording (Supplementary Movie M1). Briefly, given a video sequence acquired with a fixed camera, the background extraction method generates background scene by removing parts of the image based on motion detection. Through exclusion of pixels that appear to be moving, the algorithm establishes pixel values which do not change as potentially belonging to the background. We then adopted our bee annotation tool (https://github.com/oist/DenseObjectAnnotation) to annotate center points of each brood cell in the generated background images. We used this tool to label three initial background images from recordings L1–4 (Supplementary Fig. 2) resulting in 12 labeled images with a total of 6139 brood cells. We also annotated three images of the L5 recording containing 2397 brood cells for testing.

Based on position labels we generated segmentation labels, in which pixels within circles of radius of 10 pixels around each position label were marked as foreground. The U-Net segmentation network[20] was trained to reproduce these segmentation labels using Adam optimizer with base learning rate of 0.0001. We applied a weighting scheme in which a 2D gaussian multiplied by 10 was centered over each foreground pixel patch. Such a weighting was designed to compensate for the class imbalance between the foreground and background pixels and allowed the accurate detection of the centers of each segmentation label. The network was trained for 1000 epochs with batch size = 16.

**Position matching algorithm**. For the short-term recordings, we devised a position matching procedure linking object detections in consecutive video frames into object trajectories. We used both position coordinates as well as the posture categories full-bees and cell-bees. In the first step, all detections are considered as trajectories of length one forming the initial set of assembled trajectories $T = \{T_i : i = 1..n\}$. In each following step, detections in consecutive video frames are considered as potential extensions of the trajectories $T$. In step $i$ we calculate the Euclidian distances of the last position of each of the assembled trajectory to detections in the frame at time $t_i$. For a given trajectory $T_j$ with its last position at timepoint $t_j$ only detections below a given distance cutoff $c_d$ are considered as a match. We define the cutoff as: $c_d = a\sqrt{t_i - t_j}$ if more than 5 of the last 10 positions in a trajectory are a full-bee and $c_d = \frac{a}{3}$ otherwise, where $\alpha$ is half of a bee longest dimension, that is 40 px in our recordings. For all detections below this cutoff an additional length factor is added to the Euclidian distance:

$$l = A\left(1 - \frac{|T_j|}{\max\left(\{|T_i| : i = 1..n\}\right)}\right), \quad (1)$$

where $A = 30$ is a scaling factor chosen based on matching accuracy. The length factor prioritizes adding detections to longer trajectories instead of short trajectories that might arise as an effect of false positive detections.

After all pairs between detections in frame at time $t_i$ and last positions of the assembled trajectories are found and their distances calculated, the matchings are generated in the incremental order of distances until no more matched pairs are found. Matched detections are added to the respective trajectories. Unmatched detections in frame at time $t_i$ are added as potential starting points of new trajectories. Unmatched trajectories are kept until the time between the last detection in that trajectory and the current time exceeds a predefined gap cutoff. We define the gap cutoff as 10 sec if more than five of the last 10 positions in a trajectory are classified as cell-bee, 1 sec if the last detection in that trajectory is close to the hive entrance, and 3 sec otherwise. This choice of cutoffs is based on the observations that bees inside of the honeycomb cells can be occluded for long timespans and that large densities and fast motion near the entrance can lead to wrong matching if longer gaps are allowed. Assembled trajectories that exceed the gap cutoff are considered as finished and stored for further analysis if their length is above 1 min and discarded otherwise.

We parallelized the matching procedure by splitting the short-term recordings into segments of 1 min. Results of matching within these segments are then matched based on the criteria above. With this parallelization our approach can scale to recordings of arbitrary length with lower computational cost.

**Visual features learning**. To improve the accuracy of the trajectory reconstructions, we devised a method to exploit the visual features of bee detection images via a CNN architecture Inception V3[52]. This architecture was previously shown to perform well in the bee recognition task[49], here we altered it by adding triplet loss in the objective function[69,70]. The triplet function was originally designed for learning vector embeddings capturing similarity among entities. In the tracking context it is a function where a correct (positive) matching of detections is compared to an incorrect (negative) one. During training, triplets of bee detections are used as input including: (1) anchor image: bee detection in a frame at a timepoint $t$, (2) positive match: the same bee detection in frame at a timepoint $t + \Delta t$, (3) negative match: a different bee in the frame at a timepoint $t + \Delta t$. The objective function penalizes representations that set the positive match farther apart in terms of the Euclidian distance between the vector embedding of the images (1) and (2) than the distance of the negative match between vector embeddings of the images (1) and (3). To ensure separation of the positive and negative matches a margin $\alpha$ is added to the loss:

$$L(i_1, i_2, i_3) = \max\left(\|f(i_1) - f(i_2)\|^2 - \|f(i_1) - f(i_3)\|^2 + \alpha, 0\right) \quad (2)$$

where $i_1, i_2, i_3$ are the input images (1), (2), (3), $f$ is the embedding and the margin $\alpha = 0.5$. The loss value in a training batch is defined as mean loss of all triplets over the number of correct triplets with $L = 0$.

We estimate the number of possible triplets in one beehive of ~1000 to amount to $2.4 \times 10^8$, a number high enough to prohibit training the network within a reasonable time. Therefore, we implemented two elements in the training procedure to accelerate the learning process. First, the sampling of image triplets is done according to the criteria of the position matching algorithm described above. For a given anchor image only those negative matchings are generated that lie within the time and space distance limit to the anchor image as defined by the criteria of the matching procedure. Corresponding positive matching image is selected from the same video frame as the negative one. Second, in each step of the training, input triplets that show a positive value of the triplet loss are fed back into training. All other input triples are randomly sampled according to the rules above.

We tested a range of dimensionalities of the image embeddings (from 16 to 4096) and chose 64 as the dimensionality offering the best performance in trajectory matching, assessed as a proportion of correctly reconstructed reference trajectories, described below. We also included several data augmentation procedures including random 90° rotations and mirror random flip along both axes. Augmentation procedures were randomly sampled once for an entire triplet preserving orientation relationship among images in the original triplet. Finally, we implemented background masking to train a model capturing visual features of the bee only, excluding background (Supplementary Fig. 15). We evaluated the performance of each solution in video of hive S5 which was not included in train set.

Training comprised 5000 epochs of 128,000 batches with batch size = 32. The value of the loss function and the number of incorrect matchings in each batch did not significantly decrease beyond this number of epochs. The network was implemented in TensorFlow[93], trained with Adam optimizer[94] using a base learning rate of 0.0001. The network was initialized randomly and no additional dataset was used for validation. We compared the network performance based on tracking results.

**Matching procedure with visual features**. Quantitative representations of visual features of bee detections were integrated in the trajectory matching procedure as follows. Trajectories were assembled via analogous sequential video frame processing applying the same time and space distance cutoffs as in the position-based approach described above. For a given trajectory $T_j$ composed of detections $\{p_1 .. p_n\}$, detections in the following video frame below the distance cutoff $c_d$ to $p_n$ are considered. For each detection $d_i$ the visual similarity to trajectory $T_j$ is quantified as:

$$V = \min\left(\|f(p_j) - f(d_i)\|^2 : j = n..n - 10\right) \quad (3)$$

where $f$ is vector embedding. Detections with $V < c_v$ where $c_v = 1.75$ is a cutoff for the appearance similarity, are considered as potential extensions of $T_j$.

For all detections below the distance and appearance similarity cutoffs the distance between $T_j$ and a detection $d_j$ is defined as

$$D = BE + V + l \quad (4)$$

where $E$ is the Euclidian distance between last trajectory position $p_n$ and $d_i$, $l$ is the length factor as described above, and $B = 0.033$ is a scaling factor chosen based on accuracy of the assembled trajectories. In each step of the matching process, matching of trajectories to detections is done in and increasing order of $D$. The matching procedure exploiting visual features follows the same time gap logic and parallelization as the matching based on positions only.

For additional exploration and interpretation of the visual embedding space we implemented approaches that include orientation angle in the matching procedure.

The difference in orientation angle of detections is quantified as $d_a = \sin\left(\frac{\alpha - \alpha}{2}\right)^2$. In the "Position+angle" approach $d_a$ is added to the Euclidian distance between detection positions. In the "Embedding+angle" approach, $d_a$ is added to the $D$ of Eq. (4). In the "Rotated embedding+angle" approach, the embeddings are calculated for images of detections rotated to orient them to 0°, thus removing the orientation information from the embeddings, and $d_a$ is added to the $D$ of Eq. (4) as well. In each approach we scale $d_a$ by a factor of 0.25, which was selected as the one resulting in best performance. Results of these analyses are shown in Supplementary Fig. 17.

**Training dataset**. To obtain the initial set of trajectories for network training, we applied the previously devised "pixel personality"-based method[49] on recordings S1–S4 and the results were manually validated by inspecting videos where individual trajectories are marked. A "correct" trajectory is defined as a temporally-contiguous set of registered positions in which the same organism was identified > =80% of the timespan of a recording by manual inspection. Trajectories which are shorter than the recording length which correspond to bees that have entered or left the hive are considered correct as long as they track the bee throughout her entire presence in the hive. The proportion of correctly tracked bees is quantified relative to the mean number of bee detections in a recording.

Correct trajectories were next used as the 'initial dataset' for training the network quantifying visual feature embeddings described above. This procedure was repeated for the initial and final datasets of recordings S1–S4 and once for the recording S5. Incorrect trajectories were discarded. Due to the 80% correctness requirement, we used for network training only video segments between minutes 1 and 4 of the full 5-min-long trajectories.

The trained network was used to infer the vector embeddings of bee detections in videos S1–S4, which were then used to construct a new set of trajectories. These trajectories were again manually validated and the correct ones together with the 'initial dataset' formed the 'final dataset' of trajectories.

The network trained on the expanded train set was used to derive quantitative representations of bee detections in videos S1–S4 as well as video S5. Images from video S5 were not used in the training of the detection or the representation learning networks. Trajectories were next constructed based on the visual representations derived via this network and the results were manually validated.

**Reporting summary**. Further information on research design is available in the Nature Research Reporting Summary linked to this article.

## Data availability
All data used in our study are available[95]. The authors declare that the data supporting the findings of this study are available within the paper and its supplementary information files. Source data are provided with this paper.

## Code availability
We provide code and code tutorials[96] (https://github.com/kasiabozek/bee_tracking) and detection labeling tool (https://github.com/oist/DenseObjectAnnotation).

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

## Acknowledgements

We are grateful to Takahashi Ikemiya for maintaining the experimental bee colonies. We thank Michael Iuzzolino, Dieu My Thanh Nguyen, Orit Peleg, and Michael Smith for comments on the manuscript and code testing. This work was supported by the Okinawa Institute of Science and Technology Graduate University. Additional funding was provided by KAKENHI grants 16H06209 and 16KK0175 from the Japan Society for the Promotion of Science to AM.

## Author contributions

Y.P. performed the bee work, Y.P. and A.M. devised the imaging setup, L.H. devised the labeling tool, K.B. performed method development and data analysis, K.B., A.M., and G. S. designed the study, K.B. and G.S. wrote the manuscript.

## Funding

## Competing interests

The authors declare no competing interests.
