## [Peer Review File · Nature Communications]

Reviewers' Comments:

Reviewer #1:

Remarks to the Author:

Summary

The authors adapted a previously published convolutional neural network (CNN) for detecting honeybees inside the hive to new video recordings, and trained another CNN to identify brood cells. The former was furthermore combined with (i) a CNN that was trained to generate an embedding that can be used to distinguish individual honeybees and (ii) a detection linking method to create software that is able to simultaneously track hundreds of individuals inside the crowded confines of a honeybee hive. Bozek et al. used their two detectors to monitor population size and brood counts in 5 honeybee hives containing more than 1,000 bees over the course of months at a low temporal resolution and report a negative correlation between the two variables. In addition, they showed that their tracking software is able to follow almost 80% of the individuals in these colonies for 5 minutes, if they are recorded at a high temporal resolution.

Evaluation

Tracking hundreds of similar-looking, unmarked individuals that frequently touch or occlude each other is notoriously difficult. By adapting facial recognition techniques to distinguishing honeybees, the authors identified an elegant and computationally efficient potential solution to this problem. Unfortunately, their implementation does not include a means to re-identify an individual, which existing marker-less tracking algorithms are able to do (e.g. Ref. 34). Consequently, the time period of which individuals can be tracked is well below the duration of a typical experiment with honeybee colonies, and the identity of individuals that leave and return to the hive or have been lost track of for other reasons cannot be recovered. These limitations make the author's approach less attractive than it otherwise could be. Furthermore, the description of the method and its performance evaluation are incomplete. In addition, as the authors state, this method has already been published elsewhere. For these limitations and reasons, this manuscript is not recommended for publication in Nature Communications.

Major comments

86: The main contribution of this manuscript is an interesting method, and indeed, the authors state that "our emphasis here has been on the techniques of detection and tracking." The results should reflect this focus, yet there are more paragraphs about biological results than there are about results related to the method itself. As a true methods paper, and to give potential users a better idea of utility, there should be more information about the evaluation of the method itself.

88-104: As the authors state at the end of this paragraph, this method has already been published elsewhere. The paragraph should therefore be substantially shortened and merged with the next paragraph, and that the method has been published should be mentioned at the beginning.

118: The authors should state upfront that the performance evaluation of their brood cell detector was evaluated on its training data set (which they do mention in the Methods) and not, as it is customary and more appropriate, on an independent test data set.

125: The negative correlation between population size and brood counts relates to a fairly extensive literature on honeybee population demographics, with earlier contributions by Ratnieks, Free, Sakagami and others. None of this is cited. It is reviewed in basic texts such as Winston and Page, also not cited.

171: Because the main contribution of this manuscript is a tracking method and the way detections are linked has important implications for the capabilities of this method, the trajectory linking method should at least be briefly described and its limitations (individuals cannot be re-identified after a certain amount of time) and the consequences thereof (the number of tracked individuals decays rather quickly, no re-identification of bees returning from foraging trips) be mentioned in the Results and explicitly treated in the Discussion.

184, 215: The computational cost of the tracking method proposed in this manuscript should be measured to enable the reader to gauge whether this is fast enough to be used for real-world experiments. It should also be and compared against existing tracking solutions to substantiate the author's claim that it is low.

208, 215: The validation procedure for generating the trajectories that were used to train the CNN

is not described in the Methods. Were incorrect trajectories simply discarded or were they also corrected. If the latter, how was this correction performed?

228: If, as the authors suggest, the CNN (partially) uses the background to compute the vector embedding then it would jointly identify the bee's location in the hive and the bee itself. If this were the case, this is worth further exploration and discussion. Moreover, the authors should then refer to the embedding as a joint bee/location embedding, not an embedding that encodes the similarity between bees. An alternative interpretation of this result, namely that the background masking procedure introduces artifacts, such as partially cropped abdomens, should also be considered.

234: The generalization result is arguably the most important descriptor of the tracking method. As such, it would be helpful to know what the authors mean by "correctly tracked." Are they referring to % bees tracked or % bee detections tracked? Were individuals tracked from the beginning of the video to the end or from whenever they appeared to the end? It would be interesting to know how the length of the (partial) trajectories was distributed, what fraction of bees was not tracked, and what fraction was misidentified on a per-frame basis. These measures have important implications for determining what kinds of analyses can be performed with data generated by this tracking method.

336-389: Much, if not all, of this section refers to the segmentation method that the authors have published previously. Indeed, parts of this material are a verbatim copy of their earlier papers. The authors should shorten this section, and focus on those aspects of the method that are new, if any, or required to understand subsequent Methods sections.

424-435: The training of this CNN is described in insufficient detail. For example, how many segmentation labels were generated / how many images did the training data set encompass, how was the untrained CNN initialized, and which algorithm and parameters were used to train it? In addition, the apparent lack of a post-processing procedure that takes advantage of the fact that brood cells typically persist over many days is regrettable. The authors could add to the Discussion a few sentences about how such a procedure could improve the quality of the data generated by an automatic brood cell detector.

498: The training of this CNN is described insufficiently. For example, of how many image triples did the image library consist, how was the image library divided into training and validation data sets, and how was the CNN initialized? In addition, the name of the CNN architecture that was trained should be stated explicitly.

Minor comments

23: The authors should clarify that not the trajectories, but the segmentation results (i.e. anonymous bee and brood detections) were used to establish the negative correlation.

25: Reading the abstract for the first time, we expected that the individual behaviors were detected automatically. Instead, the authors examine some automatically generated trajectories manually and assign them to behaviors. That behaviors were identified manually should be made clear.

26: It is not obvious "what behaviors related to comb activity" means. Moreover, the only behavioral component that the authors obtain automatically is movement, which barcode-based systems monitor with ease. The statement that the identified behaviors are "difficult to detect in tagged systems" is therefore incorrect.

28: An introduction to a marker-less tracking paper for a social insect seems incomplete without references to Tucker Balch's pioneering work on marker-less tracking in honeybees and ants and the author's own published work on this topic.

45: Ref. 10 tagged all, not most individuals.

51: Partial occlusion or blur do not necessarily mean that tag recognition fails. Many barcode-based tracking systems incorporate preprocessing steps to handle motion blur (e.g. Ref. 10) and/or encode error detection and error correcting information on the barcode that permit identification even if the barcode is partially obscured (e.g. Ref. 10, Mersch et al. 2013).

78: Figure 1A and the legend suggest that a single CNN was used to detect bees and brood cells, which was not the case. The figure would also be more intuitive if arrows were to point from the tracking/detection output to the plots that were generated from this output.

103: Supplemental Table T1 is confusing. It needs a legend that clarifies what the abbreviations

mean. TP and FP suggest that the table shows true/false positive counts, yet the table shows rates in these columns (perhaps the TPR and FPR). It is unclear what "class" means or what the difference between axis and orientation is. For error measurements where this is possible, a measure of the variance should be shown to give the reader an impression of the method's precision. The row label "our method" should be changed to credit the authors of the manuscript where the bee detector was originally published.

105: It should be clarified whether this sentence refers to an assessment that was published elsewhere or not. If the latter, the results need to be shown.

109: FPR and FNR should be given as rates, not percent, to stay in line with the rest of the manuscript. Recall can be omitted, as it can easily be calculated from the FNR that the authors provide; $\text{recall} = 1 - \text{FNR}$.

110: This statement would carry more weight if at this point in the manuscript the reader already knew that some of the recordings with which the network were retrained were acquired under different conditions than S5 and L5.

116: Figure 2E does not provide sufficient information about the CNN architecture. For example, the direction of information flow is not apparent and it is unclear what the numbers mean. In addition, it is confusing that the same marker was used for bees in cells (see Figure 2C) and brood cells, because when looking at an image or movie, it would be impossible to know which of the two entities the marker represents. Other images and the actual videos use yellow circles for bees in cells, which should also be done here.

121-124 (and elsewhere): It seems like the authors use the terms "population (size)" and "(number of) detections" interchangeably. The former refers to the number of colony members that is alive while the latter is the number of colony members that was detected. Using both terms interchangeably is confusing and needs to be avoided throughout.

123: Supplemental Figure S4 is referred to before Supplemental Figure S3. The top panel of Figure S4 appears to be a duplicate of Figure 3C.

124: The authors refer to multiple peaks, but Figure 3C shows only one obvious peak. A definition of what constitutes a peak should be added to the methods, and indications of where the peaks are should be added to the figure.

125: Figure 3D doesn't show fluctuations (i.e. a measure of change over time), but simple counts. The sentence should therefore read "we found a strong negative correlation between population and brood counts". Neither the main text nor the SI state how the correlation was measured.

132: It is not obvious what the "clear phase" is that the authors refer to in the caption of Figure S7.

133: Figure S9 shows temporal cross-correlation results, but the Methods don't mention how the cross-correlation analysis was performed. Colony L3 is missing from the right panel and the legend should indicate what the dashes mean.

143: Which method was used to measure this correlation?

144: Figures S10 and S11 show that most of the time the mean distance between bees inside cells and the nearest brood cells is too big to regulate brood temperature, which honeybees are able to do in a targeted way. The authors should discuss this result in with respect to their thermoregulation hypothesis. An alternative and perhaps more parsimonious hypothesis is that at night there are simply more bees inside the hive and that the distance between bees and brood decreases because of the higher number of bees present.

166-185: These two paragraphs are not about results. They could perhaps be moved to the Introduction.

192: The caption of Fig. 1B should mention that the lines connecting bee images in the middle plot represent Euclidian distances to aid in its interpretation.

196: At this point the reader would benefit from a flow chart that outlines the training method that the authors allude to. Additionally, it would be appropriate to explicitly refer the reader to the Methods section.

203-205: The text mentions the number of organisms, but Fig. 4A shows the number of detections. The terms detection is conflated with "number of organisms," population size, and colony size throughout the manuscript, which makes it difficult to evaluate how well the method actually works (i.e. what fraction of the individuals is successfully tracked, which is important to

those who might be interested in using this method for their experiments).

209: It is not clear whether the feature embeddings were extracted from the CNN that was used to form the initial data set or whether they come from the CNN trained with triplet loss.

215: The authors should clearly state that they now use the triplet loss-trained CNN to reanalyze the training videos to make the text easier to understand.

218: Neither Fig. 4C nor Figs. S13 and S14 show the position only-based results referred to on this line. Fig. S15 shows position only results, but it is not clear with which of the two triplet loss-trained networks they were generated.

219: Since the pixel-based approach already generates up to 70% of the trajectories produced by the triplet loss-trained CNN, saying that "some trajectories ... were part of the training set" is a gross understatement. It would be more appropriate to state that most trajectories were part of the training.

235: Table S3 should show the number of bees alive to aid in the interpretation of the results presented in it.

261: This sentence should clarify that these inspections and the grouping of individuals exhibiting similar behaviors was performed manually.

277: In the context of collective behavior, a group of 100 individuals is not necessarily large. With barcode-based methods, groups that are an order of magnitude bigger than 100 have been tracked. It would therefore be more appropriate to substitute "large" for "larger."

282: Since this method automatically tracks the identity and location of individuals, but not specific behaviors," it expands marker-less tracking, not "behavioral tracking" to one large and dense collective system.

283: The authors could be more explicit about how their approach is complementary to existing ones by discussing some scenarios where their method is more suitable than others. For example, it could be used to track social insects to which no barcode can be applied because the insects are too small or because they have no hard cuticle and/or molt (termites). More generally, some of the existing barcode-based tracking methods aren't cited here (Ref. 10) or anywhere else (e.g. Blut et al, 2017, Mersch et al. 2013), and Tucker Balch et al.'s work on marker-less tracking for honeybees and ants was not mentioned, as noted previously.

407-411: It is not clear what was done here and why it was done. What is "the same approach?" What kind of predictions were generated and how precisely where they corrected and used? Is this an application of transfer learning to adapt the bee segmentation CNN different recordings? If so, the authors should mention this, state why it was necessary, and describe the image library that was used in more detail (e.g. how many image it consisted of), and lines 404-416 should be a separate subsection.

419: In addition to citing the relevant papers, the name of the background extraction method and what it does should be briefly mentioned so the reader doesn't have to read these two papers to understand how background extraction was performed.

423: Figure S3 is about background masking, not about background extraction or the labeling of brood cells.

431: These performance results have already been reported in the main text.

599: The legend should state what the error bars in panels A and C represent.

602: The horizontal order images should be reversed to better match the caption or vice versa.

612: For a researcher interested in applying this tracking method for experiments, it would be more informative to show the fraction of tracked bees in panel C instead of the fraction of tracked detected bees to include detection performance in the evaluation of the tracking algorithm (which works off the detection component's results).

616: The caption should describe what the colored dashes on the x-axis of the plots in panels A and B represent.

625: The caption should describe what the colors represent. Are these counts? Readers with a background in honeybee biology would also be interest to know where the hive entrance, because the bees' dance floor is usually close to it.

S4, S5: The y-axis labels are missing.

S6: An explanation of the abbreviations PSD and V should be added to the caption.

S9: This figure might need a more nuanced interpretation. The phase shifts are approximately 1,

6, 8, 8, and 16 hours, which is not "roughly 8 hours". There is no "bottom plot".

S10: It would be easier to visually verify the author's interpretation of these graphs if nighttime hours were shown with a darker plot background.

S11: This figure could be combined with Fig. S10.

S13: This legend is analogous to the legend of Fig. 4C, not 3C.

S16: The caption should describe what the colored dashes on the x-axis of the plots represent.

Reviewer #2:

Remarks to the Author:

The paper presents a new method to track the development of a honey bee colony and suggests to also being capable to detecting behaviours. I have read the paper with interest. In general I find the English fine, however, I wonder about the use of the word occlusion. I think you may mean obstruction of view? I have marked that in most places in the attached document.

I am not an expert on Neural Networks, and therefore I have concentrated more on the biological aspects of the introduction, results and discussion, the method parts I have limited insight into.

The idea of following a colony, using image analysis is fine. You have chosen a novel method, that does not demand to label all bees with tags. The advantage of this are clear, however, in case you want to sample an individual with a particular behaviour, tags are advantageous. Furthermore, I do think, that you need to state more frankly, the limitation that your tool demand. The hive that is used, is a two frame one side observation hive, not a full colony.

Still the working of the tool is impressive. I find it hard to follow, and the link to the movie are not functional, how you subtract the image of the bees, to get an image of the comb behind the bees, I think you could develop that part better, in particular in relation to the long time observations.

You state that you use a 12 hour interval, which makes sense, but you give no example of how the comb beneath the bees evolves, I suspect that you only consider sealed brood cells and ignore those with egg and larvae, if so please make this more clear in the text. How about cells with sealed honey, are those differentiated from sealed brood. You may benefit from reading this paper: Alves, T.S., Pinto, M.A., Ventura, P., Neves, C.J., Biron, D.G., Junior, A.C., De Paula Filho, P.L. and Rodrigues, P.J., 2020. Automatic detection and classification of honey bee comb cells using deep learning. Computers and Electronics in Agriculture, 170, p.105244.

In figure 1 I find it hard to see the details, on the small images. The third dimension is B is unnecessary and should not be used.

In figure 3 I am surprised that the colonies are so different, do the curves correspond to the exact same days? If not the illustration is confusing, but if it does, you should offer some explanation as to the asynchronous patterns of the colonies. I also find for the none biologist, it would be useful to explain that the fluctuations occur, because the bees can exit the hive to forage in the free, in other word that low number of visible bees at midday is due to fouragers leaving and reentering the hive. D the plot showing that more bees are correlated with less brood, I think is not a useful illustration. I believe it is an artefact of your setup, maybe the bees did not cope well in the observation hive.

Figure 4, again the 3D does not work well ever, and especially not at this scale.

In figure 5 D,E,F you have used different colours. However, I would find it more useful if you had coloured the three individuals differently so that in the combined fourth panel it would be simpler to see the similarities.

In summary I am happy to see that new technologies are being brought to the study of honey bees. You do offer the code for those interested, however, I think you should give more information regarding the setup of the hive, especially the instalment of light, in order to avoid reflections. You give very few details in that regard.

I find that a few additional quotes are needed, particular in your introduction, which I have added to the annotated document I attach here.

Reviewer #3:

Remarks to the Author:

Overall assessment

This manuscript communicates a significant advance toward automated understanding of individual- and colony-level behavior of a highly social insect. The work is well-motivated and presents impressive tracking results of an entire honeybee colony. This challenge seems nearly impossible at first, given that bees look very similar and are constantly maneuvering around/under/next to each other in a very crowded scene. Two features distinguish this paper from recent works in the field of animal pose estimation and tracking. First, similar to idtracker (Refs 33, 34, 46) they perform tracking of many individuals at once, but they learn a powerful representation for visual similarity instead of using a hand-crafted one. Second, this work creatively addresses the tracking problem using this visual representation; the tracking problem has not been addressed by recent deep learning methods for animal pose estimation [17, 18, Graving et al. 2019], which do not tackle scenes with multiple individuals. Beyond comparisons to recent works, the major, unique advantage is that the authors 1) develop a novel method and architecture, 2) very effectively demonstrate its real application to assessing the behavior of a complex system, and 3) provide open code, reproducible results, and a tutorial material for using their method.

Although solving the tracking problem by associating detections across frames using a combination of hand-defined (position, velocity, orientation) and learned features (visual similarity) is not a new idea within the computer vision community, the ultimate impact of many research contributions comes from the new applications they enable. This work uses state-of-the-art methods, modifies them to better suit the biological tracking domain, and importantly translates these methods, results, and datasets to the biological community. I anticipate intense interest in this work by researchers from biomechanics, entomology, and behavioral ecology.

With a few changes and additional computational analyses, I think the manuscript will be ready for publication.

Experiments

I really only have one major suggestion. I want to emphasize that I think this work is a significant contribution as-is, and would be well worth publishing even if the following suggestions do not "work out". I think there are a few additional experiments that could really help support the paper's claim that learning a visual feature embedding improves tracking of markerless animals. My basic questions are the following: Is the visual embedding capturing details of appearance beyond body orientation? Would the system perform just as well if orientation was used directly instead of visual features? In a way, body orientation is a visual feature, but what I am curious about are pose-independent visual features that capture variation in shape or texture (e.g. presence of pollen in the pollen baskets on the legs). The method already has access to the orientation data, so the following experiments might not be too difficult to implement unless computational resources are no longer available to the authors.

First, I would add two to three more "experimental conditions" in S15, and also report them in the results. Experiment 1: form trajectories using position + orientation (similar to the "Position only" condition, but with orientation angle too). Experiment 2: form trajectories using position + orientation + visual features, but with modified triplet input where each image in a triplet is pre-rotated so that the long axis of the body is along the x-axis (thus removing all orientation cues when learning the visual embedding). Optional Experiment 3: form trajectories following line 514

($D = BE + V + I$), but also include orientation. These three experiments would allow an easy comparison of how much learned visual features help beyond hand-crafted/classical features (position and orientation). Before starting on the above experiments, it may be helpful to look at a t-SNE, UMAP, or PCA plot of all bee visual embeddings (using the reported method) colored by orientation angle (to get a sense of how much the “visual features” capture orientation).

Additionally, given the potential importance of the visual features for markerless tracking, it would be interesting to see a figure presenting a visual exploration of the embedding space (e.g. plot the distribution of bee embeddings for a large sample of detections and then show example images from any clusters that arise. E.g. Do bees walking on the hive (dorsal view) and glass (ventral view) project to different regions of the embedding space? Do bees with and without pollen in their pollen baskets project to different regions?

Implementation

I am a little uncertain regarding the procedure used to predict body orientation angle. The network predictions are obviously good, so I think the issue must be my own lack of understanding.

However, I would encourage the authors to provide a discussion about the design choices I bring up here, which might assist future readers who have similar questions. I do not fully understand how the orientation loss function L [line 380] will promote good predictions. What is the rationale behind the equation for the orientation loss function L (line 380)? The equation would penalize $\hat{a} - a = 0$ the same as $\hat{a} - a = \pm 180$ (i.e. both would return $L = 0$). So *it would end up not penalizing a prediction that is the complete opposite orientation of the ground truth, and indeed would provide a gradient toward an even more incorrect answer* if the initial prediction was more than 90 degrees off). Don't you need this output in order to “assign back and front to the region principal axis” (lines 386-387)? How does the network learn from this loss except to align with the axis prediction? I see in Ref 35 that a similar loss, $\sin(a_i - a_j)$ is used when constructing a similarity measure (“Since the orientation angle is based on the body axis estimation [4], the angle component of D_{ij} penalizes deviations from axis orientation rather than the head-tail orientation.”). Maybe this is an accidental carry-over from this manuscript and the authors used a different loss function for training here? In the code, the orientation loss is $L = [\sin(\hat{a} - a)]^2$

(https://github.com/kasiabozek/bee_tracking/blob/1a09aa0534a787ca17464d63a7047a5641aed4d7/detection/unet.py#L94). Finally, I am surprised that regressing directly to orientations in the range $[0, 2\pi]$ worked as well as it did given that the representation is discontinuous (very similar samples (near 0) have very different orientations (e.g. 0.1 deg and 359.9 deg). In any case, it may be helpful to try out a different loss that provides supervision in an intermediate representation space that is continuous (following Zhou et al. On the continuity of Rotation Representations in Neural Networks. CVPR 2019. DOI: 10.1109/CVPR.2019.00589).

Presentation/Organization

This paper relies on a method (“pixel personality”) posted on arXiv by similar authors (Ref 35, the third author has changed). Ref 35 presents the method used to obtain initial trajectories [Lines 207, 521-522] and as-such, is a major component of the overall system presented here. However Ref 35 is simply referenced without much additional description, leaving the reader wondering how these initial trajectories were obtained. *This may be a decision for the Editor or authors*, but I think that this paper may want to include a summary of the method in Ref 35 (since it is a preprint), and could incorporate/communicate the main elements/methods. I believe self-plagiarism would also not be an issue if the authors updated the manuscript on arXiv (Ref 35) with this paper as a “new version”.

The word “population” has specific meaning in biology/ecology (usually the total number of individuals in a given area, and is almost always an unknown variable unless all individuals are counted through a survey). In this case it would be the underlying number of individuals or “colony size”. The way you use it here seems to be more like the “number of visible bees”. Since you are not guaranteed to see all bees in the colony at any one time I would not use “population” in this

way and would change "population" to "visible bees" throughout the manuscript.

Improving the clarity of the procedure. Just a suggestion and not a demand - I got a little mixed up in the Recordings section of the Methods [L390-416]. I guess the complexity arises because you are trying to transfer the previous method to the new domain. One thing that could improve the clarity of your overall method (e.g. for someone who is trying to replicate for a different species) is to provide a diagram of the procedural steps involved with obtaining the trained pipeline. So you might have 1) collect and densely annotate high resolution sequence (D1, D2), 2) rescale D1 and D2 to lower resolution and train detection and orientation network 3) predict detections on up to 5 initial frames and manually correct them to create a small dataset for the new, lower-resolution domain [L408-411], 4) fine-tune on this small low-res dataset for 10 iterations, etc.

In the discussion or introduction, it would also be interesting hear more about known metrics of colony health that could be obtained from individuals' tracks once they are available - e.g. for Bumblebees, activity, nursing behavior, distance from nest center, and degree of centrality all decline significantly following exposure to a Neonicotinoid pesticide [Crall et al. 2018. Neonicotinoid exposure disrupts bumblebee nest behavior, social networks, and thermoregulation. *Science* Vol. 362, Issue 6415, pp. 683-686 DOI: 10.1126/science.aat1598]

Specific comments (line number: comment). Comments, suggestions, or questions I think are particularly important are bolded.

64: Add Graving, JM, ... Couzin, ID. 2019. DeepPoseKit, a software toolkit for fast and robust animal pose estimation using deep learning. *eLife* 2019;8:e47994 DOI: 10.7554/eLife.47994

74 long-term vs short-time, maybe short-term? It would be good to use consistent terms throughout the manuscript.

96: "central parts of the bees' bodies" - it looks like this location is near the petiole (segment between the abdomen and thorax) - a reference to Fig. 2A might help here.

101: change "allowed to" to "allowed us to"

101: Consider merging text "with a precision of ..." with the accuracy results in paragraph lines 105-112.

101: " a precision of" - I think you might mean position error here? Are you reporting the distribution of errors? If so, I would have interpreted precision as 1/variance, but would be more interested in the mean abs. error (distance of predictions from ground truth). Readers may get tripped up because TPR and FPR are discussed immediately after. Precision is also a well-known metric ($TP/TP+FP$) for measuring detection performance at a given threshold (usually average precision across a range of thresholds is reported).

106-107: "retraining on a set of up to five frames" - do you mean fine-tuning?

108: "that were not included in the retraining" - Do you mean "which were not included during retraining"? Do you mean to remind the reader that L5 and S5 were not in the training set? Using "that" could lead the reader to believe there are also frames from the beginning, middle and end parts that were included in the retraining.

108-109: "estimated detection FPR... 0.98" - How are these different from L101-104 in the paragraph above? I think these results should be merged into a single paragraph if they are different, or one version should be eliminated if they refer to the same evaluation.

116-117: "We used round-shaped ... with no overlap" - How do you identify clusters of pixels in the segmented image (e.g. blob detection?) and how do you enforce no overlap between the centers?

118: "FNR ~ 0.05" - In Supp. Movie M2, there seem to be many filled cells along the outer edges of the central concentration (e.g. $t=6-10$ sec). Are these brood cells (i.e. are they false negatives), or do they contain honey or pollen instead? What is happening with the cluster of what looks like filled cells to the right ($t = 41$)? Bees are not visiting these cells and they are not being predicted as brood cells. Later at $t = 49$ sec they appear to be replaced with slightly brighter caps, some of which are newly classified as brood cells. In addition, it seems like the background result for M1 is much cleaner than for M2 - any idea why this is?

119: long-term or long-time?

121: "total population" - In biology, the population usually describes the underlying number of individuals or "colony size". The way you use it here seems to be more like the "number of visible bees". Since you are not guaranteed to see all bees in the colony at any one time I would not use "population" in this way and would **change** "population" to "visible bees" throughout the manuscript. A related question: What fraction of bees are outside of the hive at any point in time? Can you use the visual features from your tracker to estimate the true number of bees in the colony?

123: "period of these fluctuations" - Presumably this is the result of bees going out and foraging? If so, what is going on with L3 and L4? Why does the "population" peak during the day? Is this real or is the tracking system just failing to detect them during this time period? How does detection performance break down across full- and cell-bees? If performance for cell bees is worse, this could explain the drop in "population" during the night. Oh I see, explained in the next paragraph.

124-125: "Over longer times, peaks in population followed peaks in brood numbers" - Is there an associated statistical statement or analysis supporting this claim? Maybe a figure similar to S9, but with visible bees - brood number?

126-127: "perhaps an indication of homeostatic control of colony size" - Is there a biological reference for this? How would a negative correlation support the "homeostatic control" hypothesis vs. an alternative hypothesis?

129-130: "We found high numbers ... sleeping bees" - Although I think this is indeed obvious from the figure (except for L5), please consider adding supporting statistics using a circular analysis of variance (e.g. with the package 'circular' in R) including hive as a 'group'. Same for results in Figure S7.

133-135: "two colonies (L3 and L4) were filmed during particularly high temperatures..., which can encourage bees to stay at the hive entrance" - Hive L2 also had high temperatures, but does not appear to exhibit the same phenomena.

144-145: "phase relationship suggests thermoregulatory activities ... brood temperature at night" - It would be nice to have a reference here. Is this a known behavior or method of maintaining brood temperature?

156-157: "The cell-bees in those hives were additionally located much farther away", Figure S10 - **The minimum mean distance to brood for L5 is higher than the maximum mean distance for any other hive!** Any explanation? Are they just ignoring their brood? Or did brood detection fail? Or was there a pixels-to-meters calibration error?

187: "distinct pixel fingerprints" - Do you mean the pixels in the image? Or do you mean the "pixel personality" mentioned above, which seem to be learned appearance features as well? A little more emphasis on definitions and consistent use of terms here would be good.

193: "is the massive space" - Is the issue that it is hard to choose which triplets are best/informative for training? Perhaps include a reference describing this hurdle, if available.

197: "in time and space which are sources of potential identity swaps" - Is this meant to read (adding a comma) "in time and space, which are sources..." or do you mean that triplets were sampled mostly from moments where potential identity swaps occurred?

208: "correct trajectories" - Are these trajectories for the full 5 min? Without going and reading Ref 35, the reader should be able to get a sense of how extensive these trajectories are. Is a trajectory correct if there are no ID swaps and it is longer than 1 min, or is it only correct if it follows the target bee over the entire set of 9,000 frames? What is the distribution of trajectory length for the reference trajectories? Maybe it can be guessed given that ~90% of detections are assigned to tracks and that the number of reference trajectories is less than the mean number of simultaneously detected bees. See also my note for Lines 522-523.

218: "a position only-based solution" - Which figure is this shown in? Do you mean Figure S15?

227-228: "data augmentation...increases data complexity hindering network convergence" - I understand that some augmentation procedures may result in training images that are outside the domain of the true data, but I am surprised that rotation and reflection have this large of a negative effect. Because positive samples are sampled from nearby frames, it seems reasonable that reflection and large rotation would disrupt the embedding since the orientation of the bee is probably one of the primary features captured by the embedding. Trying to bring together encodings from the same bee in wildly different orientations while also separating embeddings

from different individuals would indeed be a more challenging problem. Is there any indication that visual features of bees are not symmetric under reflection, in which case only rotation augmentation would be appropriate unless all three images in the triplet were reflected in the same way? See also my note for Line 495.

228-230: "Poorer results...background ...suggest it plays an important role..." - In addition to providing background features at the bee's location, the background may also contain parts of other bees bodies, which could also help with matching.

234-235: "We found that 77% of bees were correctly tracked (Fig. 4C)" - I think this statement slightly misconstrues what Fig 4C shows. The trajectories capture 77% of detected bees, which include ~96% of true bees.

243: "Movies M9-M13" - These numbers do not line up with groupings on <https://groups.oist.jp/bptu/honeybee-tracking-dataset>, which do you mean here?

246-248: "distributions of individuals' speed...trajectory" - I see the distributions of mean speed and angular speed in Figure 5 and motion span and diffusion presented in S16. Additional statistics about these trajectories should be presented here or in Figure S16. I would include the distributions of trajectory duration (number of frames) and path length (e.g. summed distances between observations along a trajectory) or possibly a scatterplot of both together with marginal densities to the side and top. My interest in seeing these distributions is to 1) see the distribution of trajectory length in the dataset, and 2) understand how many of the bees are just sitting there, which would change the difficulty of the tracking problem. Additionally, would it be possible to calculate statistics on the distribution of occlusions across tracks and how many of these are successfully handled by the tracking system without ID swaps? This would give a great sense of the overall level of difficulty, and the range of difficulty among different tracks. If not, no big deal.

252: "24% of the high diagonal on average" - change "high" to "hive". The hive surface is 47 cm x 47 cm, with a maximum diagonal of $47 \cdot \sqrt{2} \sim 66 \text{ cm} = 660 \text{ mm}$. How is the diagonal of the surface motion getting to be higher than 660 mm - up to 2500 mm in some cases (Figure S16)? Is "diagonal of the surface of motion" actually some form of path length? I might replace this part of Fig S16 with path length anyway. See comment for lines 246-248.

254: "indicative of the foragers performing waggle dance" - Stepping through one of the waggle dance videos frame by frame (M13) it does not seem like the frame rate or exposure are fast enough to capture the actual waggle of the dance. That said, I would think that the repeating reorientation during the dance would be fairly easy to detect from the orientation data from the trajectory. If you want to make this waggle dance statement about colonies S2 and S5, I would be more convinced by # waggle dances (maybe normalized by population) per time period rather than the distribution of mean angular speed over the entire trajectory, which to me seems like could be more related to walking speed and having to turn around often to stay within the hive. I think demonstrating detection of actual waggle dances from the angle data or some spatial analysis (e.g. Fig. 5D) could also serve to further demonstrate the value of this system as providing real metrics of social activity.

262-263: "Trajectories with high angular and translational motion indeed belong to ... waggle dance" It would be great to see solid statistics on this. See my comment for line 254. It seems you might be able to quickly annotate what fraction of these high angular and translational motion trajectories do in fact have waggle dances. I understand this is meant to be an exemplar of the kind of behavioral analysis one could do with the trajectory data, but I think the paper would be more powerful if you complete the connection with a concrete behavior and analyze statistically.

265: "M14-M16" - On the website these are M13-M15?

269: "M17-M19" - M16-28 on the website.

287: "uncapped honey [cells] as well as pollen cells"? (add cells?)

348-351: "We also submitted a subset ... estimate of human errors... labeling task." - How did you determine the False Positive Rate of human annotations? Similar to thresholding confidence scores of a classifier to move along the tradeoff between precision and recall, for the 10 labelers, where did you set the threshold level of agreement for which you would count a labeled position as a true positive/negative? E.g. if two labelers clicked where nobody else did, it's probably a false positive (but could also be 8 false negatives). If four labelers didn't click where

everyone else did, it's probably four false negatives, but could easily be 6 false positives.

362-364, 380: "defined as $L = \sin((\hat{a} - a)/2)$... predicted and labeled orientation angle, respectively" - I am surprised that regressing directly to orientations in the range $[0, 2\pi]$ worked as well as it did given that the representation is discontinuous. Additionally, **what is the rationale behind the equation for the orientation loss function L** [line 380]? The equation would penalize $\hat{a} - a = 0$ the same as $\hat{a} - a = \pm 180$ (i.e. both would return $L = 0$). So *it would end up not penalizing a prediction that is the complete opposite orientation of the ground truth*, and indeed would provide a gradient toward an even more incorrect answer if the initial prediction was more than 90 degrees off). Don't you need this output in order to "assign back and front to the region principal axis" [lines 386-387? How does the network learn from this loss except to randomly align with the axis prediction? Maybe I am missing something major because the network predictions are obviously good. In any case, it may be helpful to try out a different loss following Zhou et al. On the continuity of Rotation Representations in Neural Networks. CVPR 2019. DOI: 10.1109/CVPR.2019.00589.

385: "axis full-bee" - should be "axis for full-bee"

386: "a given regions" - should be "a given region"

407: "We retrained the same approach..." - Do you mean you trained the network from Ref 13 and 35 from scratch on the rescaled images? "Retrained" makes it sound like the first training was part of the method for this paper (maybe it is?).

411: "We then retrained the detection model" - Do you mean fine-tuned? In general I have been a little confused in this section about what you mean when you say retrained.

438-439: "We used both position coordinates as well as the posture categories..." - I think the authors may want to try also adding an orientation matching cost. If performance is still lower than when using visual features, it would serve as a powerful demonstration that the visual feature embedding is capturing something in addition to orientation. See main comments above.

448: in the denominator inside the $\max(\cdot)$, is " $i-1..n$ " supposed to be " $i=1..n$ "?

449-450: "The length factor prioritizes longer trajectories" - this might be more clear if you rephrase to "prioritizes [adding new detections] to longer trajectories" (if this is correct)

455-456: "Unmatched trajectories are kept until the time between ... and the current time is below ..." - I think you mean "current time is above..."?

481-482: "over the number of correct triplets with $L = 0$ " - Where is this $L/\text{num_correct}$ coming from? Is this meant to encourage larger updates for batches where there were not very many correct triplets (i.e. hard batches)? What if there are no correct triplets in the batch?

490: "are fed back into training" - Do you mean they are added directly to the next batch?

495: "augmentation procedures ..." - Was each image within a triplet randomly rotated/flipped separately? I think **augmentation via a single randomly chosen rotation/flip that is then applied simultaneously to all three images in the triplet might improve performance** because it wouldn't introduce as much of a domain gap between training and augmented distributions. If relevant visual features are asymmetric, flipping images independently within a triplet could hurt performance. Similarly rotating the anchor and positive separately might also decrease performance if the main visual feature that is being extracted is the orientation of the bee. See comments for lines 227-228.

514: " $D = BE + V + I$ " - I think a few additional experiments replacing V with an orientation angle distance could really help demonstrate the value of matching using visual features over just orientation. See main comments above.

522: "**manually validated**" - Please describe the procedure for manual validation including number of validators that scored each potential trajectory. For example, this could be an outlier, but M16_tr1_P17_421_highincell_2764.17_4.00.mp4 contains an ID swap at $t = 57$ seconds. The target bee from preceding frames is occluded as several other bees walk over it and then continues to the left after the other bees walk over it. On the other hand, M17 shows a very impressive track recovery at $t = 1$ min 45 sec. I think these swaps have relatively little impact on overall training with the triplet loss, but their presence should be noted or at least estimated. In fact, these swaps could potentially be located by looking for spikes in D [line 514], calculated for each frame along a trajectory.

522-523: "**Correct trajectories**" - Please define what it means for a trajectory to be

correct. From what I gather, it means a trajectory that is more than one minute long with no ID swaps?

529: "The network trained on this way expanded train set" - remove "way"

530: "Images of video S5" - maybe should be "Images from video S5"?

530: "were not used in training neither" - I don't think you want a double negative here. Maybe rephrase to "were not used in training either the detection or representation learning networks"?

590-591: (Figure 3E) "High numbers of cell-bees occur predominantly in the evening..." - Do you have any information whether these bees are "sleeping" or might they be feeding brood?

613: (Figure 4C) "an earlier approach" - This probably isn't feasible, but it would be great to a comparison with idtracker (refs 33, 34, 46).

625: (Figure 5C) "The spatial distribution of trajectories characterized by large linear and angular motion." - What does color measure? The number of observations for each position bin?

Minor asks: Add equation numbers throughout unless not consistent with journal formatting.

Supplement

Figures S4 and S5. The meaning of black and gray lines for raw and smoothed values appears to have switched? It would be great if they were consistent. Also, please add a description of these to the figure legend.

Figure S7. please consider adding supporting statistics using a circular analysis of variance (e.g. with the package 'circular' in R) including hive as a 'group'.

Figure S9. change "bell-bee" to "cell-bee"

Figure S9. It could be more informative to have time shift range from -12 h to 12 h, which would make the high brood number plot more interpretable (i.e. the orange vertical line would be to the left of the green vertical line).

Supp Table 1: Why does Supp T1 not have an entry for human true positive rate? Is it 1.0? See my comment for lines 348-351. Also, the "Axis" column measurement is not defined anywhere in the manuscript. This is the error of the principal component direction of the network segmentation, correct? "Orientation angle" would be the error after assigning head/tail based on mean orientation pixel values for the region? It could be helpful to add these definitions to the table legend.

T3 description, third line: Change "The trajectories constructed by each tracking methods are..." to "The trajectories constructed by each tracking method are..." (methods to singular).

We thank the reviewers for the extensive and in-depth evaluation of our manuscript. We appreciate the enormous effort and the thoughtfulness of the comments and the suggested improvements. The revised manuscript contains 14 new and revised figures, 18 new citations, and a considerably improved method and results description. We hope that the revised manuscript and additional analyses have sufficiently addressed the concerns of the reviewers and achieved a quality appropriate for publication in *Nature Communications*.

REVIEWER COMMENTS

Reviewer #1 (Remarks to the Author):

Summary

The authors adapted a previously published convolutional neural network (CNN) for detecting honeybees inside the hive to new video recordings, and trained another CNN to identify brood cells. The former was furthermore combined with (i) a CNN that was trained to generate an embedding that can be used to distinguish individual honeybees and (ii) a detection linking method to create software that is able to simultaneously track hundreds of individuals inside the crowded confines of a honeybee hive. Bozek et al. used their two detectors to monitor population size and brood counts in 5 honeybee hives containing more than 1,000 bees over the course of months at a low temporal resolution and report a negative correlation between the two variables. In addition, they showed that their tracking software is able to follow almost 80% of the individuals in these colonies for 5 minutes, if they are recorded at a high temporal resolution.

Evaluation

Tracking hundreds of similar-looking, unmarked individuals that frequently touch or occlude each other is notoriously difficult. By adapting facial recognition techniques to distinguishing honeybees, the authors identified an elegant and computationally efficient potential solution to this problem. Unfortunately, their implementation does not include a means to re-identify an individual, which existing marker-less tracking algorithms are able to do (e.g. Ref. 34). Consequently, the time period of which individuals can be tracked is well below the duration of a typical experiment with honeybee colonies, and the identity of individuals that leave and return to the hive or have been lost track of for other reasons cannot be recovered. These limitations make the author's approach less attractive than it otherwise could be. Furthermore, the description of the method and its performance evaluation are incomplete. In addition, as the authors state, this method has already been published elsewhere. For these limitations and reasons, this manuscript is not recommended for publication in *Nature Communications*.

We thank the Reviewer for the thorough and critical reading of our manuscript. The Reviewer correctly points out that our method does not currently allow for recognition of bees coming in and out of the hive. A robust and automatic inference of individual identity over long timescales from videos is an ultimate aim of our machine vision tracking approach. Indeed, the techniques and the dataset presented in our manuscript are foundational for further development of identity determination and we have no doubt that the tracking timespan as well as the long-term recognition capacity of our solution will improve. In our study, we choose the tracking time interval both to enable full manual validation of the results (itself a massive task) and because colony-wide behavior at the resolution of an individual organism is still very interesting (and scientifically unexplored) within these intervals. We note that many tracking methods present accuracy estimation on simulated data or only partially validated results (e.g. Belsch et al. 2001, Newby et al. 2018, Liu et al. 2018, cited in our manuscript).

In order to improve the method description in the revised manuscript among other corrections we:

- performed systematic tests of accuracy of brood detection and bee detection in independent test sets (described in detail in the response below),
- explicitly state the limitations of the tracking method,
- describe in detail the trajectory validation procedure,
- clarify our methods for estimation of tracking completeness used in the manuscript,
- added a new Supplemental Fig. S11 to illustrate the training procedure as well as added technical details about network training,
- show the distribution of trajectory and path lengths in the new Supplemental Fig. S18.

Finally, it was our intention to be transparent about the previous publication of the detection technique in a conference proceedings, a fact we mentioned in our cover letter. The detection represents only a limited part of the work presented in this manuscript and the brood detection as well as the tracking methods are all quite novel. The results, observations and the annotated tracking dataset are presented uniquely in this manuscript.

Major comments

86: The main contribution of this manuscript is an interesting method, and indeed, the authors state that “our emphasis here has been on the techniques of detection and tracking.” The results should reflect this focus, yet there are more paragraphs about biological results than there are about results related to the

method itself. As a true methods paper, and to give potential users a better idea of utility, there should be more information about the evaluation of the method itself.

In the revised manuscript we expanded the method description by:

- performing systematic tests of accuracy of brood detection and bee detection in independent test sets (Results, l. 118-119, 126-129, Supplemental Fig. S1-S2, Table T1),
- describing in detail the trajectory validation procedure and clarifying our methods for estimation of tracking completeness used in the manuscript and the validation procedure (l. 544-550),
- adding a new Supplemental Fig. S11 that illustrates the training procedure and adding technical details about the network training.

We have also made text modifications throughout the manuscript.

88-104: As the authors state at the end of this paragraph, this method has already been published elsewhere. The paragraph should therefore be substantially shortened and merged with the next paragraph, and that the method has been published should be mentioned at the beginning.

We shortened this paragraph, merged it with the paragraph below and added text referencing our previous method (l. 101-121 in the revised manuscript).

118: The authors should state upfront that the performance evaluation of their brood cell detector was evaluated on its training data set (which they do mention in the Methods) and not, as it is customary and more appropriate, on an independent test data set.

This is an important remark, thank you. We have quantified brood detection performance on a set of frames of the L5 recording that were not part of the training set (l. 126-129) The detected brood cells are also shown in Supplemental Fig. S2 and Supplemental Movie M2.

125: The negative correlation between population size and brood counts relates to a fairly extensive literature on honeybee population demographics, with earlier contributions by Ratnieks, Free, Sakagami and others. None of this is cited. It is reviewed in basic texts such as Winston and Page, also not cited.

We thank the reviewer for pointing us to this literature. We have included the suggested works in the revised manuscript (l. 142).

171: Because the main contribution of this manuscript is a tracking method and the way detections are linked has important implications for the capabilities of this method, the trajectory linking method should at least be briefly described and its limitations (individuals cannot be re-identified after a certain amount of time) and the consequences thereof (the number of tracked individuals decays rather quickly, no re-

identification of bees returning from foraging trips) be mentioned in the Results and explicitly treated in the Discussion.

In the revised version of the manuscript we explicitly state the limitations of the tracking method as suggested by the Reviewer (Introduction l. 94-95, Discussion l. 308-311).

184, 215: The computational cost of the tracking method proposed in this manuscript should be measured to enable the reader to gauge whether this is fast enough to be used for real-world experiments. It should also be and compared against existing tracking solutions to substantiate the author's claim that it is low.

This is an important point that is however difficult to address due to differences in computational hardware used for ours and previously published methods. The detection and tracking procedure for each of our 5 min recordings takes ~1 h on a typical 4-GPU compute node which we explain in the revised version of the manuscript (l. 220-221). A previously published tracking method, Wario et al. 2015, used a Cray XC30 supercomputer, featuring 117 TiB of RAM and 1872 compute nodes with 24 CPU cores each (total of 44928 processors) to process a 9-weeks long recording.

208, 215: The validation procedure for generating the trajectories that were used to train the CNN is not described in the Methods. Were incorrect trajectories simply discarded or were they also corrected. If the latter, how was this correction performed?

We did not alter any trajectories. Incorrect trajectories were identified and removed based on visual inspection. We explain this procedure in more detail in the revised version of the manuscript (Methods l. 544-550).

228: If, as the authors suggest, the CNN (partially) uses the background to compute the vector embedding then it would jointly identify the bee's location in the hive and the bee itself. If this were the case, this is worth further exploration and discussion. Moreover, the authors should then refer to the embedding as a joint bee/location embedding, not an embedding that encodes the similarity between bees. An alternative interpretation of this result, namely that the background masking procedure introduces artifacts, such as partially cropped abdomens, should also be considered.

We now refer to the embedding as 'detection image embedding' throughout the manuscript. We also reformulated the interpretation of the results in cropped images as potentially pertaining to the cropping artifacts (l. 229-231 in the revised manuscript). We additionally performed analysis of the contribution of the orientation angle and posture to the embedding, as well as added an exploration of the embedding space via clustering that we present in new Supplemental Figures S12 and S13. The additional figures reveal that the embeddings combine orientation, posture and background information.

234: The generalization result is arguably the most important descriptor of the tracking method. As such, it would be helpful to know what the authors mean by “correctly tracked.” Are they referring to % bees tracked or % bee detections tracked? Were individuals tracked from the beginning of the video to the end or from whenever they appeared to the end? It would be interesting to know how the length of the (partial) trajectories was distributed, what fraction of bees was not tracked, and what fraction was misidentified on a per-frame basis. These measures have important implications for determining what kinds of analyses can be performed with data generated by this tracking method.

We refer to the percentage of detections tracked, where the number of detections is averaged across the video frames. A “correct” trajectory is defined as a temporally-contiguous set of registered positions in which the same organism was identified $\geq 80\%$ of the time span of a recording by manual inspection. Trajectories which are shorter than the recording length or which correspond to bees that have entered or left the hive are considered correct as long as they follow a bee throughout her entire presence in the hive. These clarifications have been added to the revised version of the manuscript l. 544-550, as well as in the legend of the Supplemental Table T3.

We added a plot of distribution of trajectory lengths in time and space as a new Supplemental Fig. S18. The accuracy of detection was estimated in two independent recordings L5 and S5 and provided in the initial and expanded in the revised version of the manuscript (l. 118-119). We also updated Supplemental Fig. S1 and Supplemental Table T1 to better illustrate the detection results.

336-389: Much, if not all, of this section refers to the segmentation method that the authors have published previously. Indeed, parts of this material are a verbatim copy of their earlier papers. The authors should shorten this section, and focus on those aspects of the method that are new, if any, or required to understand subsequent Methods sections.

We shortened this section according to the Reviewer’s suggestion (l. 354-380).

424-435: The training of this CNN is described in insufficient detail. For example, how many segmentation labels were generated / how many images did the training data set encompass, how was the untrained CNN initialized, and which algorithm and parameters were used to train it? In addition, the apparent lack of a post-processing procedure that takes advantage of the fact that brood cells typically persist over many days is regrettable. The authors could add to the Discussion a few sentences about how such a procedure could improve the quality of the data generated by an automatic brood cell detector.

We added the suggested details in the revised version of the manuscript. We also conducted a more thorough performance evaluation in an independent test set of frames from video L5 (l. 127-129). As noted by the Reviewer, we did not use the temporal component in this network. We added this

information as a comment in the Discussion (l. 291-293). The capped cells are visually distinct enough such that additional postprocessing was unnecessary for a small potential gain in accuracy above the already achieved of TPR ~ 0.99 , FPR ~ 0.01 , FNR < 0.01 in the test set.

498: The training of this CNN is described insufficiently. For example, of how many image triples did the image library consist, how was the image library divided into training and validation data sets, and how was the CNN initialized? In addition, the name of the CNN architecture that was trained should be stated explicitly.

We have added the requested information to the revised version of the manuscript (l. 478, 493, 508-509). We did not include the number of triplets in the train set as the triplets were generated during training and their number was never explicitly estimated. We have also performed the following calculation. For a beehive of size $\sim 1,000$ organisms imaged over 2,400 video frames, a bee in a video frame could have ~ 100 possible wrong matches. If we additionally include different Δt that we use in the training, the number of possible triplets amounts to 2.4×10^8 for one beehive and hence could reach $\sim 10^9$ in our entire train set.

Minor comments

23: The authors should clarify that not the trajectories, but the segmentation results (i.e. anonymous bee and brood detections) were used to establish the negative correlation.

We reworded this sentence (l. 21-22).

25: Reading the abstract for the first time, we expected that the individual behaviors were detected automatically. Instead, the authors examine some automatically generated trajectories manually and assign them to behaviors. That behaviors were identified manually should be made clear.

These behaviors were automatically filtered and manually validated. The automatic behavior filtering was possible based on the motion parameters that we register via our tracking approach (speed, angular speed, being in a comb cell). The fact that our motion parameters allow for such efficient behavior filtering holds the promise that with a larger dataset where more instances of these behaviors exist, a more systematic detection approach can be developed. We reworded this formulation in the revised version of the manuscript (l. 25-26) and also added statistical evaluation that motion parameters correspond to certain behaviors (l. 265-268).

26: It is not obvious “what behaviors related to comb activity” means. Moreover, the only behavioral component that the authors obtain automatically is movement, which barcode-based systems monitor with

ease. The statement that the identified behaviors are “difficult to detect in tagged systems” is therefore incorrect.

By “difficult to detect in tagged systems” we mean behaviors which occlude the tag, such as crawling inside of a comb cell. Crawling inside of a cell does not disrupt tracking in our solution which we now described explicitly in the revised version of the manuscript (l. 26).

28: An introduction to a marker-less tracking paper for a social insect seems incomplete without references to Tucker Balch’s pioneering work on marker-less tracking in honeybees and ants and the author’s own published work on this topic.

We thank the reviewer for this suggestion. We added Tucker Balch’s work to the introduction of the revised version of the manuscript.

45: Ref. 10 tagged all, not most individuals.

We corrected the text accordingly (l. 45).

51: Partial occlusion or blur do not necessarily mean that tag recognition fails. Many barcode-based tracking systems incorporate preprocessing steps to handle motion blur (e.g. Ref. 10) and/or encode error detection and error correcting information on the barcode that permit identification even if the barcode is partially obscured (e.g. Ref. 10, Mersch et al. 2013).

We corrected this formulation according to the reviewer’s suggestion (l. 54).

78: Figure 1A and the legend suggest that a single CNN was used to detect bees and brood cells, which was not the case. The figure would also be more intuitive if arrows were to point from the tracking/detection output to the plots that were generated from this output.

We thank the reviewer for this suggestion. We reformulated the legend and altered the figure.

103: Supplemental Table T1 is confusing. It needs a legend that clarifies what the abbreviations mean. TP and FP suggest that the table shows true/false positive counts, yet the table shows rates in these columns (perhaps the TPR and FPR). It is unclear what “class” means or what the difference between axis and orientation is. For error measurements where this is possible, a measure of the variance should be shown to give the reader an impression of the method’s precision. The row label “our method” should be changed to credit the authors of the manuscript where the bee detector was originally published.

We edited the legend and the table for clarity.

105: It should be clarified whether this sentence refers to an assessment that was published elsewhere or not. If the latter, the results need to be shown.

Here we refer to the previously published work. We have shortened and clarified the entire section.

109: FPR and FNR should be given as rates, not percent, to stay in line with the rest of the manuscript. Recall can be omitted, as it can easily be calculated from the FNR that the authors provide; $\text{recall} = 1 - \text{FNR}$.

We thank the reviewer for pointing out this mistake, which we have corrected (l. 118).

110: This statement would carry more weight if at this point in the manuscript the reader already knew that some of the recordings with which the network were retrained were acquired under different conditions than S5 and L5.

We thank the reviewer for this suggestion and we added this information (l. 119).

116: Figure 2E does not provide sufficient information about the CNN architecture. For example, the direction of information flow is not apparent and it is unclear what the numbers mean. In addition, it is confusing that the same marker was used for bees in cells (see Figure 2C) and brood cells, because when looking at an image or movie, it would be impossible to know which of the two entities the marker represents. Other images and the actual videos use yellow circles for bees in cells, which should also be done here.

We added more clarifications to the caption of this image. Here as well as in other images throughout the manuscript that illustrate detection accuracy, we used yellow color to mark labels and red to mark predictions. In panel 2C those two colors are highly overlapping, for this reason, the markers appear red. Due to this coloring convention we would prefer to keep the originally proposed colors in this image.

121-124 (and elsewhere): It seems like the authors use the terms “population (size)” and “(number of) detections” interchangeably. The former refers to the number of colony members that is alive while the latter is the number of colony members that was detected. Using both terms interchangeably is confusing and needs to be avoided throughout.

It is an important remark. We changed the term “population” to “visible bees” in the revised manuscript as the actual population size is not known to us. We use term “detection” for the detection method description. We checked and tightened this naming in the revised version of the manuscript.

123: Supplemental Figure S4 is referred to before Supplemental Figure S3. The top panel of Figure S4 appears to be a duplicate of Figure 3C.

We thank the reviewer for the thorough reading of the manuscript. We corrected supplemental figures order and removed the redundant panel of Figure S4.

124: The authors refer to multiple peaks, but Figure 3C shows only one obvious peak. A definition of what constitutes a peak should be added to the methods, and indications of where the peaks are should be added to the figure.

We clarified the statement to make it clear that in this example colony there is one clear and dominant peak (l. 138-139).

125: Figure 3D doesn't show fluctuations (i.e. a measure of change over time), but simple counts. The sentence should therefore read "we found a strong negative correlation between population and brood counts". Neither the main text nor the SI state how the correlation was measured.

We are using the word "fluctuations" with its statistical meaning. We changed this sentence accordingly and added correlation information (l. 139-142).

132: It is not obvious what the "clear phase" is that the authors refer to in the caption of Figure S7.

We removed this term from the sentence (l. 146) as we did not quantify the phase (now Supplemental Fig. S6).

133: Figure S9 shows temporal cross-correlation results, but the Methods don't mention how the cross-correlation analysis was performed. Colony L3 is missing from the right panel and the legend should indicate what the dashes mean.

This is Figure S8 in the revised manuscript. We expanded the figure legend to better explain the cross-correlation and dashes. Colony L3 did not show sufficient brood counts and was hence not included in the right panel. We added this explanation to the figure legend.

143: Which method was used to measure this correlation?

It is the Pearson correlation and we added this information in the revised version of the manuscript (l. 159).

144: Figures S10 and S11 show that most of the time the mean distance between bees inside cells and the nearest brood cells is too big to regulate brood temperature, which honeybees are able to do in a targeted way. The authors should discuss this result in with respect to their thermoregulation hypothesis. An alternative and perhaps more parsimonious hypothesis is that at night there are simply more bees inside the hive and that the distance between bees and brood decreases because of the higher number of bees present.

We removed the thermoregulation hypothesis from the manuscript as it is insufficiently supported by our data. The fact that cell-bees tend to be located close to brood cells is however supported by our data and we don't consider this is due to the limited size of our beehives. In none of the

recordings is the space completely filled and there are always empty comb cells present (l. 160-163).

166-185: These two paragraphs are not about results. They could perhaps be moved to the Introduction.

We moved this content to the introduction in the revised manuscript (l. 68-82).

192: The caption of Fig. 1B should mention that the lines connecting bee images in the middle plot represent Euclidian distances to aid in its interpretation.

This is a very helpful remark and we added this explanation to the figure legend.

196: At this point the reader would benefit from a flow chart that outlines the training method that the authors allude to. Additionally, it would be appropriate to explicitly refer the reader to the Methods section.

We added a flow chart describing the training procedure as a new Supplemental Fig. S11. We also refer the reader to the Methods section in the revised version of the manuscript (l. 197).

203-205: The text mentions the number of organisms, but Fig. 4A shows the number of detections. The terms detection is conflated with “number of organisms,” population size, and colony size throughout the manuscript, which makes it difficult to evaluate how well the method actually works (i.e. what fraction of the individuals is successfully tracked, which is important to those who might be interested in using this method for their experiments).

We corrected and tightened the naming in the revised version of the manuscript. The detection and tracking accuracy are now also more extensively quantified (Figures S1-S2, Table T1, Results 1. 118-119, Methods 1. 400-402) which should help future users to assess the expected precision of our method.

209: It is not clear whether the feature embeddings were extracted from the CNN that was used to form the initial data set or whether they come from the CNN trained with triplet loss.

We thank the reviewer for this comment, we reformulated this piece of text (l. 205-206).

215: The authors should clearly state that they now use the triplet loss-trained CNN to reanalyze the training videos to make the text easier to understand.

We reformulated this sentence according to the reviewer’s suggestion (l. 213-214).

218: Neither Fig. 4C nor Figs. S13 and S14 show the position only-based results referred to on this line. Fig. S15 shows position only results, but it is not clear with which of the two triplet loss-trained networks they were generated.

Supplemental Figure S14 in the revised manuscript shows comparison with pixel personality approach, Supplemental Figure S15 with the position only approach. We corrected the figure numbering and added the missing information to the legend of Supplemental Figure S15.

219: Since the pixel-based approach already generates up to 70% of the trajectories produced by the triplet loss-trained CNN, saying that “some trajectories ... were part of the training set” is a gross understatement. It would be more appropriate to state that most trajectories were part of the training.

We corrected this formulation accordingly (l. 217).

235: Table S3 should show the number of bees alive to aid in the interpretation of the results presented in it.

We did not observe any dead bees during manual validation, and we have no indication that dead bees would significantly change our findings.

261: This sentence should clarify that these inspections and the grouping of individuals exhibiting similar behaviors was performed manually.

We added this information to this paragraph (l. 264).

277: In the context of collective behavior, a group of 100 individuals is not necessarily large. With barcode-based methods, groups that are an order of magnitude bigger than 100 have been tracked. It would therefore be more appropriate to substitute “large” for “larger.”

We corrected the text accordingly (l. 282).

282: Since this method automatically tracks the identity and location of individuals, but not specific behaviors,” it expands marker-less tracking, not “behavioral tracking” to one large and dense collective system.

We corrected this sentence accordingly (l. 287).

283: The authors could be more explicit about how their approach is complementary to existing ones by discussing some scenarios where their method is more suitable than others. For example, it could be used to track social insects to which no barcode can be applied because the insects are too small or because they have no hard cuticle and/or molt (termites). More generally, some of the existing barcode-based tracking methods aren't cited here (Ref. 10) or anywhere else (e.g. Blut et al, 2017, Mersch et al. 2013), and Tucker Balch et al.'s work on marker-less tracking for honeybees and ants was not mentioned, as noted previously.

We added complementarity explanations and the suggested citations to this paragraph (l. 290-291).

407-411: It is not clear what was done here and why it was done. What is “the same approach?” What kind of predictions were generated and how precisely where they corrected and used? Is this an application of transfer learning to adapt the bee segmentation CNN different recordings? If so, the authors should mention this, state why it was necessary, and describe the image library that was used in more detail (e.g. how many image it consisted of), and lines 404-416 should be a separate subsection.

For clarity, we have rephrased these sentences and split this section into multiple paragraphs. In short, our previously developed detection network (which outputs position and orientation estimates) needed to be retrained primarily to introduce new backgrounds into the training set. The dataset on which the method was originally developed did not contain empty parts of the comb which resulted in large number of false positives on the empty background. The network was trained from scratch first on the D1-D2 dataset then on frames of S1-S4 and L1-L4, so this is not transfer learning. We did resize the data on which the method was developed to match the resolution of videos used in this manuscript. (l. 417-426).

419: In addition to citing the relevant papers, the name of the background extraction method and what it does should be briefly mentioned so the reader doesn't have to read these two papers to understand how background extraction was performed.

The background extraction method is based on motion detection. We added explanatory sentences to the revised version of the manuscript (l. 430-433).

423: Figure S3 is about background masking, not about background extraction or the labeling of brood cells.

We have corrected the reference to Figure S2.

431: These performance results have already been reported in the main text.

We removed the performance information from this section.

599: The legend should state what the error bars in panels A and C represent.

We added this information to the legend in the revised version of the manuscript.

602: The horizontal order images should be reversed to better match the caption or vice versa.

We corrected this image panel caption.

612: For a researcher interested in applying this tracking method for experiments, it would be more informative to show the fraction of tracked bees in panel C instead of the fraction of tracked detected bees to include detection performance in the evaluation of the tracking algorithm (which works off the detection component's results).

Based on the quantification of the detection algorithm accuracy (in an independent test set S5, L5) we can expect detection FPR of ~ 0.03 and FNR of ~ 0.01 . These values roughly correspond to the error bars presented in this figure, which show the percentage range over the varying numbers of detections in different frames. We additionally show these results as proportions of all reference trajectories we have for these recordings (Supplemental Fig. S14). It would however be impossible to validate all detections in our data which amount to ~ 15 million.

616: The caption should describe what the colored dashes on the x-axis of the plots in panels A and B represent.

The colored dashes indicate the mean of each distribution (speed and angular speed for all 5 colonies). This is an important omission and we added this information to the legend.

625: The caption should describe what the colors represent. Are these counts? Readers with a background in honeybee biology would also be interested to know where the hive entrance, because the bees' dance floor is usually close to it.

The colors are counts for each location in the hive and we have added this information to the revised manuscript.

S4, S5: The y-axis labels are missing.

These are S3, S4 in the revised manuscript. We explain now the y-axis in the figure legend.

S6: An explanation of the abbreviations PSD and V should be added to the caption.

This is S5 in the revised manuscript. We added the explanations to the caption.

S9: This figure might need a more nuanced interpretation. The phase shifts are approximately 1, 6, 8, 8, and 16 hours, which is not "roughly 8 hours". There is no "bottom plot".

This is S8 in the revised manuscript. We corrected this caption in the revised manuscript by removing our interpretation. As the correlation values are small for majority of the hives the phase agreement is not very strongly supported.

S10: It would be easier to visually verify the author's interpretation of these graphs if nighttime hours were shown with a darker plot background.

This is S9 in the revised manuscript and we shaded nighttime hours.

S11: This figure could be combined with Fig. S10.

This is S9 in the revised manuscript and we have combined the earlier Fig. S10 and S11.

S13: This legend is analogous to the legend of Fig. 4C, not 3C.

This is S14 in the revised manuscript and we corrected the caption accordingly.

S16: The caption should describe what the colored dashes on the x-axis of the plots represent.

This is S19 in the revised manuscript and we have added this information to the caption.

Reviewer #2 (Remarks to the Author):

The paper presents a new method to track the development of a honey bee colony and suggests to also being capable to detecting behaviours. I have read the paper with interest. In general I find the English fine, however, I wonder about the use of the word occlusion. I think you may mean obstruction of view? I have marked that in most places in the attached document.

We thank the reviewer for the insightful response to our manuscript. We have changed the word “occlusion” with “visual occlusion” throughout the manuscript. We have also included all the suggestions listed in the document attached by the reviewer.

I am not an expert on Neural Networks, and therefore I have concentrated more on the biological aspects of the introduction, results and discussion, the method parts I have limited insight into. The idea of following a colony, using image analysis is fine. You have chosen a novel method, that does not demand to label all bees with tags. The advantage of this are clear, however, in case you want to sample an individual with a particular behaviour, tags are advantageous. Furthermore, I do think, that you need to state more frankly, the limitation that your tool demand. The hive that is used, is a two frame one side observation hive, not a full colony.

We thank the reviewer for this comment. In the revised manuscript we have included more extensive discussion of the limitations of our method, including the one-side observation beehive, as well as limitations pointed out by Reviewer 1 (l. 308-311).

Still the working of the tool is impressive. I find it hard to follow, and the link to the movie are not functional, how you subtract the image of the bees, to get an image of the comb behind the bees, I think you could develop that part better, in particular in relation to the long time observations. You state that you use a 12 hour interval, which makes sense, but you give no example of how the comb beneath the bees evolves, I suspect that you only consider sealed brood cells and ignore those with egg and larvae, if so please make this more clear in the text. How about cells with sealed honey, are those differentiated from sealed brood. You may benefit from reading this paper: Alves, T.S., Pinto, M.A., Ventura, P., Neves, C.J., Biron, D.G., Junior, A.C., De Paula Filho, P.L. and Rodrigues, P.J., 2020. Automatic detection and classification of honey bee comb cells using deep learning. *Computers and Electronics in Agriculture*, 170, p.105244. In figure 1 I find it hard to see the details, on the small images. The third dimension is B is unnecessary and should not be used.

We have included the background extraction method in the revised manuscript (l. 430-433). We also fixed the links to the supplemental movies which are illustrative of how the background evolves in time. We indeed consider only capped brood in the analysis and we make this point more

explicitly in the revised version of the manuscript (l. 295-296). We also cite and discuss the study suggested by the reviewer (l. 129-132). Figure 1B was also corrected.

In figure 3 I am surprised that the colonies are so different, do the curves correspond to the exact same days? If not the illustration is confusing, but if it does, you should offer some explanation as to the asynchronous patterns of the colonies. I also find for the none biologist, it would be useful to explain that the fluctuations occur, because the bees can exit the hive to forage in the free, in other word that low number of visible bees at midday is due to fouragers leaving and reentering the hive. D the plot showing that more bees are correlated with less brood, I think is not a useful illustration. I believe it is an artefact of your setup, maybe the bees did not cope well in the observation hive.

Yes, panel A of Figure 3 does illustrate different hives aligned by time of the day. While there are regularities, such as the approximately 24-h period cycle shown in Fig. 3A (right) there is also substantial variability, an aspect we discuss extensively (l. 133-157 in the revised manuscript) and also an interesting topic for future work with a large number of colonies. We include the possibility of increasing the size of observation beehive as an important future development of our method (l. 309-311).

Figure 4, again the 3D does not work well ever, and especially not at this scale.

This is only illustrative of separation of the tracked bee (red color) relative to the remaining bees (yellow color). It is not a quantitative result only qualitative illustration. As the original dimension of this space is 64-dimensional, we decided to use 3D plot as symbolically representing the complexity of this space.

In figure 5 D,E,F you have used different colours. However, I would find it more useful if you had coloured the three individuals differently so that in the combined fourth panel it would be simpler to see the similarities.

We changed the coloring in these three panels.

In summary I am happy to see that new technologies are being brought to the study of honey bees. You do offer the code for those interested, however, I think you should give more information regarding the setup of the hive, especially the instalment of light, in order to avoid reflections. You give very few details in that regard.

We added the description of the hive illumination system to the revised version of the manuscript (l. 340-343).

I find that a few additional quotes are needed, particular in your introduction, which I have added to the annotated document I attach here.

We thank the reviewer for these comments, we added the suggested literature to the revised version of the manuscript.

Per Kryger

Reviewer #3 (Remarks to the Author):

Overall assessment

This manuscript communicates a significant advance toward automated understanding of individual- and colony-level behavior of a highly social insect. The work is well-motivated and presents impressive tracking results of an entire honeybee colony. This challenge seems nearly impossible at first, given that bees look very similar and are constantly maneuvering around/under/next to each other in a very crowded scene. Two features distinguish this paper from recent works in the field of animal pose estimation and tracking. First, similar to idtracker (Refs 33, 34, 46) they perform tracking of many individuals at once, but they learn a powerful representation for visual similarity instead of using a hand-crafted one. Second, this work creatively addresses the tracking problem using this visual representation; the tracking problem has not been addressed by recent deep learning methods for animal pose estimation [17, 18, Graving et al. 2019], which do not tackle scenes with multiple individuals. Beyond comparisons to recent works, the major, unique advantage is that the authors 1) develop a novel method and architecture, 2) very effectively demonstrate its real application to assessing the behavior of a complex system, and 3) provide open code, reproducible results, and a tutorial material for using their method.

Although solving the tracking problem by associating detections across frames using a combination of hand-defined (position, velocity, orientation) and learned features (visual similarity) is not a new idea within the computer vision community, the ultimate impact of many research contributions comes from the new applications they enable. This work uses state-of-the-art methods, modifies them to better suit the biological tracking domain, and importantly translates these methods, results, and datasets to the biological community. I anticipate intense interest in this work by researchers from biomechanics, entomology, and behavioral ecology.

With a few changes and additional computational analyses, I think the manuscript will be ready for publication.

Sincerely,

Marc Badger

We thank the reviewer for positive assessment and appreciation of our work. The detailed comments of the Reviewer have allowed us the opportunity to importantly improve the manuscript. We hope that with the added analyses, new figures and explanations, we have added more insights and value. Below, we present detailed answers to Reviewer's comments.

Experiments

I really only have one major suggestion. I want to emphasize that I think this work is a significant contribution as-is, and would be well worth publishing even if the following suggestions do not “work out”. I think there are a few additional experiments that could really help support the paper’s claim that learning a visual feature embedding improves tracking of markerless animals. My basic questions are the following: Is the visual embedding capturing details of appearance beyond body orientation? Would the system perform just as well if orientation was used directly instead of visual features? In a way, body orientation is a visual feature, but what I am curious about are pose-independent visual features that capture variation in shape or texture (e.g. presence of pollen in the pollen baskets on the legs). The method already has access to the orientation data, so the following experiments might not be too difficult to implement unless computational resources are no longer available to the authors.

First, I would add two to three more “experimental conditions” in S15, and also report them in the results. Experiment 1: form trajectories using position + orientation (similar to the “Position only” condition, but with orientation angle too). Experiment 2: form trajectories using position + orientation + visual features, but with modified triplet input where each image in a triplet is pre-rotated so that the long axis of the body is along the x-axis (thus removing all orientation cues when learning the visual embedding). Optional Experiment 3: form trajectories following line 514 ($D = BE + V + I$), but also include orientation. These three experiments would allow an easy comparison of how much learned visual features help beyond hand-crafted/classical features (position and orientation). Before starting on the above experiments, it may be helpful to look at a t-SNE, UMAP, or PCA plot of all bee visual embeddings (using the reported method) colored by orientation angle (to get a sense of how much the “visual features” capture orientation).

We find the suggestions of the reviewer very relevant and interesting. In response to this comment we implemented two more tracking strategies “Position+angle”, “Rotated embedding + angle”, and “Embedding+angle”. It turns out that while orientation angle improves the accuracy of matching based on position only, it does not affect matching based on embeddings. Matchings based on “Position+angle” and “Rotated embedding+angle” do not reach the accuracy of matching based on embeddings of images in their original orientation, as shown in new Supplemental Figure S17. We hypothesize this is due to the fact that the visual embeddings combine body orientation, posture and background information in a complicated manner. To further support these observations, we performed an analysis of the space of visual features described in the point below. We added these results as the new Supplemental Figures S12, S13 and S17.

Additionally, given the potential importance of the visual features for markerless tracking, it would be interesting to see a figure presenting a visual exploration of the embedding space (e.g. plot the distribution

of bee embeddings for a large sample of detections and then show example images from any clusters that arise. E.g. Do bees walking on the hive (dorsal view) and glass (ventral view) project to different regions of the embedding space? Do bees with and without pollen in their pollen baskets project to different regions?

We appreciate this suggestion from the reviewer and were happy to perform the suggested analyses. We provide example images of embedding space of bee detections included in supplemental videos M3-M7. Each of the video contains between 25,000 and 50,000 detections. We visualized the detections in 3D using t-SNE and split the embeddings into 20 clusters. 25 members of each clusters were plotted for visual inspection. While fine-scale detail, such as pollen baskets are unlikely an important factor in the embeddings, we suggest that the embeddings combine orientation angle, posture (inside cell, upside down), and background composition. We provide these results as Supplemental Figures S12 and S13, the approach is described in l. 533-541.

Implementation

I am a little uncertain regarding the procedure used to predict body orientation angle. The network predictions are obviously good, so I think the issue must be my own lack of understanding. However, I would encourage the authors to provide a discussion about the design choices I bring up here, which might assist future readers who have similar questions. I do not fully understand how the orientation loss function L [line 380] will promote good predictions. What is the rationale behind the equation for the orientation loss function L (line 380)? The equation would penalize $\hat{a} - a = 0$ the same as $\hat{a} - a = \pm 180$ (i.e. both would return $L = 0$). *So it would end up not penalizing a prediction that is the complete opposite orientation of the ground truth, and indeed would provide a gradient toward an even more incorrect answer if the initial prediction was more than 90 degrees off.* Don't you need this output in order to “assign back and front to the region principal axis” (lines 386-387)? How does the network learn from this loss except to align with the axis prediction? I see in Ref 35 that a similar loss, $\sin(a_i - a_j)$ is used when constructing a similarity measure (“Since the orientation angle is based on the body axis estimation [4], the angle component of D_{ij} penalizes deviations from axis orientation rather than the head-tail orientation.”). Maybe this is an accidental carry-over from this manuscript and the authors used a different loss function for training here? In the code, the orientation loss is $L = [\sin(\hat{a} - a)]^2$ (https://github.com/kasiabozek/bee_tracking/blob/1a09aa0534a787ca17464d63a7047a5641aed4d7/detecti on/unet.py#L94). Finally, I am surprised that regressing directly to orientations in the range $[0, 2\pi]$ worked as well as it did given that the representation is discontinuous (very similar samples (near 0) have very different orientations (e.g. 0.1 deg and 359.9 deg). In any case, it may be helpful to try out a different loss that provides supervision in an intermediate representation space that is continuous (following

Zhou et al. On the continuity of Rotation Representations in Neural Networks. CVPR 2019. DOI: 10.1109/CVPR.2019.00589).

We thank the reviewer for the thorough reading of our methods and provide our clarifications here below.

The parameter inside the sine in the loss function has a denominator equal 2. That means the range of loss is value of sine function between 0 and 180 degree where function values close to 0 result from orientation angle difference close to 0 and 360 degrees. Loss function value close to 1 results from orientation angle difference close to 180 degrees. The square power of the sine function was missing in the text and we corrected this in the revised version of the manuscript.

In the implementation the orientation angles are scaled between 0 and 1. The angle values are multiplied by π (instead of 2π) before applying sine function which is equivalent to the function described in the text.

Presentation/Organization

This paper relies on a method ('pixel personality') posted on arXiv by similar authors (Ref 35, the third author has changed). Ref 35 presents the method used to obtain initial trajectories [Lines 207, 521-522] and as-such, is a major component of the overall system presented here. However Ref 35 is simply referenced without much additional description, leaving the reader wondering how these initial trajectories were obtained. *This may be a decision for the Editor or authors*, but I think that this paper may want to include a summary of the method in Ref 35 (since it is a preprint), and could incorporate/communicate the main elements/methods. I believe self-plagiarism would also not be an issue if the authors updated the manuscript on arXiv (Ref 35) with this paper as a "new version".

Following this suggestion we included description of this method in the revised version of the manuscript (l. 76-81).

The word "population" has specific meaning in biology/ecology (usually the total number of individuals in a given area, and is almost always an unknown variable unless all individuals are counted through a survey). In this case it would be the underlying number of individuals or "colony size". The way you use it here seems to be more like the "number of visible bees". Since you are not guaranteed to see all bees in the colony at any one time I would not use "population" in this way and would change "population" to "visible bees" throughout the manuscript.

Thank you for this comment and we are sensitive to how terms can be used differently across fields. To avoid any confusion, we now use "visible bees" throughout the revised manuscript. To explain our thinking we were using "population" in the sense of "set of items under consideration", as it is

also used in the cell- or particle-tracking literature (e.g. Bruijning et al. “Automated particle tracking to obtain population counts and size distributions from videos in r” <https://doi.org/10.1111/2041-210X.12975>, Li et al. “Cell Population Tracking and Lineage Construction with Spatiotemporal Context” doi: 10.1016/j.media.2008.06.001, Lee-Ling et al. “Tracking of cell population from time lapse and end point confocal microscopy images with multiple hypothesis Kalman smoothing filters” doi: 10.1109/CVPRW.2010.5543444).

Improving the clarity of the procedure. Just a suggestion and not a demand - I got a little mixed up in the Recordings section of the Methods [L390-416]. I guess the complexity arises because you are trying to transfer the previous method to the new domain. One thing that could improve the clarity of your overall method (e.g. for someone who is trying to replicate for a different species) is to provide a diagram of the procedural steps involved with obtaining the trained pipeline. So you might have 1) collect and densely annotate high resolution sequence (D1, D2), 2) rescale D1 and D2 to lower resolution and train detection and orientation network 3) predict detections on up to 5 initial frames and manually correct them to create a small dataset for the new, lower-resolution domain [L408-411], 4) fine-tune on this small low-res dataset for 10 iterations, etc.

We extensively rewrote this section in the revised version of the manuscript (l. 417-426).

In the discussion or introduction, it would also be interesting hear more about known metrics of colony health that could be obtained from individuals’ tracks once they are available - e.g. for Bumblebees, activity, nursing behavior, distance from nest center, and degree of centrality all decline significantly following exposure to a Neonicotinoid pesticide [Crall et al. 2018. Neonicotinoid exposure disrupts bumblebee nest behavior, social networks, and thermoregulation. Science Vol. 362, Issue 6415, pp. 683-686 DOI: 10.1126/science.aat1598]

Among behaviors that could correspond to colony health are nursing, sleep, and communication signals. We added discussion of these potential metrics that could be obtained from trajectories to discussion (l. 317-319).

Specific comments (line number: comment). Comments, suggestions, or questions I think are particularly important are bolded.

64: Add Graving, JM, ... Couzin, ID. 2019. DeepPoseKit, a software toolkit for fast and robust animal pose estimation using deep learning. eLife 2019;8:e47994 DOI: 10.7554/eLife.47994

We added this citation.

74 long-term vs short-time, maybe short-term? It would be good to use consistent terms throughout the manuscript.

We changed to short-term.

96: “central parts of the bees’ bodies” - it looks like this location is near the petiole (segment between the abdomen and thorax) - a reference to Fig. 2A might help here.

We added the figure reference to this sentence. This correction occurs in sentences that were moved to the methods (l. 365) following comments of Reviewer 1.

101: change “allowed to” to “allowed us to”

We added ‘us’ to this formulation.

101: Consider merging text “with a precision of ...” with the accuracy results in paragraph lines 105-112.

These paragraphs were shortened and merged in the revised manuscript.

101: “ a precision of” - I think you might mean position error here? Are you reporting the distribution of errors? If so, I would have interpreted precision as $1/\text{variance}$, but would be more interested in the mean abs. error (distance of predictions from ground truth). Readers may get tripped up because TPR and FPR are discussed immediately after. Precision is also a well-known metric ($TP/TP+FP$) for measuring detection performance at a given threshold (usually average precision across a range of thresholds is reported).

It is an important remark – changed “precision” to “error” in this sentence. It is now in the methods (l. 400-402).

106-107: “retraining on a set of up to five frames” - do you mean fine-tuning?

We added this formulation to the sentence (l. 115).

108: “that were not included in the retraining” - Do you mean “which were not included during retraining”? Do you mean to remind the reader that L5 and S5 were not in the training set? Using “that” could lead the reader to believe there are also frames from the beginning, middle and end parts that were included in the retraining.

We rephrased this sentence to make it clear that the entire recordings were excluded from the training (l. 117).

108-109: “estimated detection FPR... 0.98” - How are these different from L101-104 in the paragraph above? I think these results should be merged into a single paragraph if they are different, or one version should be eliminated if they refer to the same evaluation.

Following this and Reviewer's 1 comment we now shortened this paragraph, merged it with the following and moved the results of the detection method on its originally dedicated dataset to the methods part (l. 400-402). We hope this presentation is clearer. The detection results are now also compiled in Supplemental Table T1.

116-117: "We used round-shaped ... with no overlap" - How do you identify clusters of pixels in the segmented image (e.g. blob detection?) and how do you enforce no overlap between the centers?

Yes, pixel patches are detected as connected foreground pixel blobs. By choosing a small size of the foreground patches we limit the possibility of overlaps between patches that mark separate objects (bees, brood cells), though some overlaps are still possible. We added this clarification to the methods part (l. 368-370).

118: "FNR ~ 0.05" - In Supp. Movie M2, there seem to be many filled cells along the outer edges of the central concentration (e.g. $t=6-10$ sec). Are these brood cells (i.e. are they false negatives), or do they contain honey or pollen instead? What is happening with the cluster of what looks like filled cells to the right ($t = 41$)? Bees are not visiting these cells and they are not being predicted as brood cells. Later at $t = 49$ sec they appear to be replaced with slightly brighter caps, some of which are newly classified as brood cells. In addition, it seems like the background result for M1 is much cleaner than for M2 - any idea why this is?

We improved and tightened the estimation of the accuracy of this method (l. 126-129). The outside cells detected as brood later in the movie potentially represent false positives as the reviewer points out. As they are outside of the central brood patch, they most probably contain food stores that rarely appear in our beehives and hence were not included in the train set of the detection method. We suspect that the blur in the background of M2 is due to the presence of bees that move very little. Background in this recording might be constantly obscured and difficult to extract. While we currently have no explanation for this phenomenon, we think it might be potentially related to the poor health of this colony.

119: long-term or long-time?

We standardized our terminology to "long-term" and corrected the sentence.

121: "total population" - In biology, the population usually describes the underlying number of individuals or "colony size". The way you use it here seems to be more like the "number of visible bees". Since you are not guaranteed to see all bees in the colony at any one time I would not use "population" in this way and would **change "population" to "visible bees" throughout the manuscript**. A related question: What

fraction of bees are outside of the hive at any point in time? Can you use the visual features from your tracker to estimate the true number of bees in the colony?

We changed “population” to “visible bees” throughout the manuscript.

We propose that an approximation of the number of bees outside the hive could be calculated as the difference between the maximum number of bees present during a 24 h or longer time span and the number of detected bees at a given time point.

123: “period of these fluctuations” - Presumably this is the result of bees going out and foraging? If so, what is going on with L3 and L4? Why does the "population" peak during the day? Is this real or is the tracking system just failing to detect them during this time period? How does detection performance break down across full- and cell-bees? If performance for cell bees is worse, this could explain the drop in "population" during the night. Oh I see, explained in the next paragraph.

We posit that the difference in detection performance is small enough not to introduce this amount of variability during 24 h. Interpretation of these results is provided in following paragraphs.

124-125: Over longer times, peaks in population followed peaks in brood numbers” - Is there an associated statistical statement or analysis supporting this claim? Maybe a figure similar to S9, but with visible bees - brood number?

This is not a quantitative result, only an observation, as there is not sufficient data to prove it. We reformulated this sentence in the revised manuscript (l. 138-139).

126-127: “perhaps an indication of homeostatic control of colony size” - Is there a biological reference for this? How would a negative correlation support the "homeostatic control" hypothesis vs. an alternative hypothesis?

We have no explanation of this phenomenon at this time, however as the effect is visible across the hive we present this result in the manuscript.

129-130: “We found high numbers ... sleeping bees” - Although I think this is indeed obvious from the figure (except for L5), please consider adding supporting statistics using a circular analysis of variance (e.g. with the package 'circular' in R) including hive as a 'group'. Same for results in Figure S7.

We added information about the circular mean value and the result of Rayleigh test of uniformity for the values plotted in these figures (Figures 3E and S6 now).

133-135: “two colonies (L3 and L4) were filmed during particularly high temperatures..., which can encourage bees to stay at the hive entrance” - Hive L2 also had high temperatures, but does not appear to exhibit the same phenomena.

The temperature was slightly lower during the filming of L2. Our temperature hypothesis is only speculative and altered activity could arise from other factors, such as those arising from environment pollutants, a point we have added to the manuscript (l. 150-151).

144-145: “phase relationship suggests thermoregulatory activities ... brood temperature at night” - It would be nice to have a reference here. Is this a known behavior or method of maintaining brood temperature?

Even though this is a known method for thermoregulation (Kleinhenz M, Bujok B, Fuchs S, Tautz J. 2003. Hot bees in empty brood nest cells: heating from within. *Journal of Experimental Biology* 206(23):4217–4231 DOI 10.1242/jeb.00680. Bujok, B., Kleinhenz, M., Fuchs, S. and Tautz, J., 2002. Hot spots in the bee hive. *Naturwissenschaften*, 89(7), pp.299-301), we removed the thermoregulation hypothesis from the manuscript as it is insufficiently supported by our data.

156-157: “The cell-bees in those hives were additionally located much farther away”, Figure S10 - **The minimum mean distance to brood for L5 is higher than the maximum mean distance for any other hive!** Any explanation? Are they just ignoring their brood? Or did brood detection fail? Or was there a pixels-to-meters calibration error?

We attribute this difference, especially at the beginning and in the second half of the recording, to the low brood counts in the hive and bees located all over the hive which increases the mean distance value. We did not detect any major detection error in this data and the measurement methods are the same for all five recordings.

187: “distinct pixel fingerprints” - Do you mean the pixels in the image? Or do you mean the "pixel personality" mentioned above, which seem to be learned appearance features as well? A little more emphasis on definitions and consistent use of terms here would be good.

We mean ‘pixel personality’ here and have reformulated this sentence (l. 181).

193: “is the massive space” - Is the issue that it is hard to choose which triplets are best/informative for training? Perhaps include a reference describing this hurdle, if available.

The problem here is the dataset size and the time it would take if we were to use all possible triplets during training. We added an explanatory sentence to the revised manuscript (l. 190-191).

197: “in time and space which are sources of potential identity swaps” - Is this meant to read (adding a comma) "in time and space, which are sources..." or do you mean that triplets were sampled mostly from moments where potential identity swaps occurred?

Yes, this is the solution we propose here. We only pick training triplets where the negative match is close in space to the positive match and could lead to errors during matching procedure. We have added a comma to this sentence as suggested by the Reviewer.

208: “correct trajectories” - Are these trajectories for the full 5 min? Without going and reading Ref 35, the reader should be able to get a sense of how extensive these trajectories are. Is a trajectory correct if there are no ID swaps and it is longer than 1 min, or is it only correct if it follows the target bee over the entire set of 9,000 frames? What is the distribution of trajectory length for the reference trajectories? Maybe it can be guessed given that ~90% of detections are assigned to tracks and that the number of reference trajectories is less than the mean number of simultaneously detected bees. See also my note for Lines 522-523.

A “correct” trajectory is defined as a temporally-contiguous set of registered positions in which the same organism was identified $\geq 80\%$ of the time span of a recording by manual inspection. Trajectories which are shorter than the recording length which correspond to bees that have entered or left the hive are considered correct as long as they track the bee throughout her entire presence in the hive. These clarifications have been added to the revised version of the manuscript (l. 544-550), as well as in the legend of the Supplemental Table T3.

218: “a position only-based solution” - Which figure is this shown in? Do you mean Figure S15?

We corrected the figure numbering and referencing in the revised version of the manuscript. We reference here Figures S14-S15 of the revised version of the manuscript.

227-228: “data augmentation...increases data complexity hindering network convergence” - I understand that some augmentation procedures may result in training images that are outside the domain of the true data, but I am surprised that rotation and reflection have this large of a negative effect. Because positive samples are sampled from nearby frames, it seems reasonable that reflection and large rotation would disrupt the embedding since the orientation of the bee is probably one of the primary features captured by the embedding. Trying to bring together encodings from the same bee in wildly different orientations while also separating embeddings from different individuals would indeed be a more challenging problem. Is there any indication that visual features of bees are not symmetric under reflection, in which case only rotation augmentation would be appropriate unless all three images in the triplet were reflected in the same way? See also my note for Line 495.

We apply the same image modification to all three images in a triplet. Once randomly sampled, the same operation is used in all three images otherwise the triplets would differ too much from real-

world examples. We clarify this in the revised version of the manuscript (l. 506-507). We would like to point out that the difference in performance is actually subtle <5% of detected bee number.

228-230: “Poorer results...background ...suggest it plays an important role...” - In addition to providing background features at the bee’s location, the background may also contain parts of other bees bodies, which could also help with matching.

We expanded this sentence in the revised manuscript (l. 229-231).

234-235: “We found that 77% of bees were correctly tracked (Fig. 4C)” - I think this statement slightly misconstrues what Fig 4C shows. The trajectories capture 77% of detected bees, which include ~96% of true bees.

We reformulated this sentence adding “detected bees” for more precision (l. 235).

243: “Movies M9-M13” - These numbers do not line up with groupings on <https://groups.oist.jp/bptu/honeybee-tracking-dataset>, which do you mean here?

We meant videos M8-M12 and this is corrected in the revised manuscript.

246-248: “distributions of individuals’ speed...trajectory” - I see the distributions of mean speed and angular speed in Figure 5 and motion span and diffusion presented in S16. Additional statistics about these trajectories should be presented here or in Figure S16. I would include the **distributions of trajectory duration (number of frames) and path length** (e.g. summed distances between observations along a trajectory) or possibly a scatterplot of both together with marginal densities to the side and top. My interest in seeing these distributions is to 1) see the distribution of trajectory length in the dataset, and 2) understand how many of the bees are just sitting there, which would change the difficulty of the tracking problem. Additionally, would it be possible to calculate statistics on the distribution of occlusions across tracks and how many of these are successfully handled by the tracking system without ID swaps? This would give a great sense of the overall level of difficulty, and the range of difficulty among different tracks. If not, no big deal.

As these distributions vary among the hives, we plotted them separately for each hive and for all hives together. They are now added as Supplemental Fig. S18 in the revised manuscript. Most of the individuals across the hives move locally, however in each of the hives we have found trajectories that span almost three times the hive dimension.

252: “24% of the high diagonal on average” - change “high” to “hive”. The hive surface is 47 cm x 47 cm, with a maximum diagonal of $47 \times \sqrt{2} \sim 66 \text{ cm} = 660 \text{ mm}$. **How is the diagonal of the surface motion getting to be higher than 660 mm - up to 2500 mm in come cases (Figure S16)?** Is "diagonal of the

surface of motion" actually some form of path length? I might replace this part of Fig S16 with path length anyway. See comment for lines 246-248.

By mistake, this figure presented distances in pixel values and was not exchanged for the corrected version before submission. The correct figure is Supplemental Fig. S19 in the revised manuscript where the distance is in mm.

254: "indicative of the foragers performing waggle dance" - Stepping through one of the waggle dance videos frame by frame (M13) it does not seem like the frame rate or exposure are fast enough to capture the actual waggle of the dance. That said, I would think that the repeating reorientation during the dance would be fairly easy to detect from the orientation data from the trajectory. If you want to make this waggle dance statement about colonies S2 and S5, I would be more convinced by # waggle dances (maybe normalized by population) per time period rather than the distribution of mean angular speed over the entire trajectory, which to me seems like could be more related to walking speed and having to turn around often to stay within the hive. I think demonstrating detection of actual waggle dances from the angle data or some spatial analysis (e.g. Fig. 5D) could also serve to further demonstrate the value of this system as providing real metrics of social activity.

Following the Reviewer's comment, we inspected 100 trajectories showing highest translational and angular motion across the 5 hives. While averaging the velocity is not the most precise measure, over the short time span of our recordings, the number of dancers is significantly related to higher average speed ($N = 10$, $p \sim 0.003$). We also observed a significantly elevated number of dancers' followers among the fast-moving trajectories ($N = 27$, $p \sim 0.025$) suggesting the chain effect of dancers elevating motion velocities of all the surrounding bees – followers and bees pushed around by their motion. For these reasons the fast angular and translation motion is ~ 2.5 times more likely to occur close to the hive entrance ($p < 0.01$). We added these statistics to the revised manuscript (l. 259-261, 265-268).

As the Reviewer points out here, we are not detecting behaviors in a systematic manner within this manuscript. We propose a set of metrics and find that they strongly correspond to certain behaviors. It will be exciting to revisit these points in detail in future work.

262-263: "Trajectories with high angular and translational motion indeed belong to ... waggle dance" It would be great to see solid statistics on this. See my comment for line 254. It seems you might be able to quickly annotate what fraction of these high angular and translational motion trajectories do in fact have waggle dances. I understand this is meant to be an exemplar of the kind of behavioral analysis one could do with the trajectory data, but I think the paper would be more powerful if you complete the connection with a concrete behavior and analyze statistically.

Please see our response to the previous comment.

265: “M14-M16” - On the website these are M13-M15?

269: “M17-M19” - M16-28 on the website.

We corrected the video numbering in the revised version of the manuscript.

287: “uncapped honey [cells] as well as pollen cells”? (add cells?)

We added “cells”.

348-351: “We also submitted a subset ... estimate of human errors... labeling task.” - **How did you determine the False Positive Rate of human annotations?** Similar to thresholding confidence scores of a classifier to move along the tradeoff between precision and recall, for the 10 labelers, where did you set the threshold level of agreement for which you would count a labeled position as a true positive/negative? E.g. if two labelers clicked where nobody else did, it's probably a false positive (but could also be 8 false negatives). If four labelers didn't click where everyone else did, it's probably four false negatives, but could easily be 6 false positives.

We used our “reference label” for the estimation of human annotators labeling. This reference label was inspected by us and corrected if any striking errors were present. We took this labeled and error-proofed portions of data as reference. We added this clarification to the revised manuscript (l. 361).

362-364, 380: “defined as $L = \sin((\hat{a} - a)/2)$... predicted and labeled orientation angle, respectively” - I am surprised that regressing directly to orientations in the range $[0, 2\pi]$ worked as well as it did given that the representation is discontinuous. Additionally, **what is the rationale behind the equation for the orientation loss function L [line 380]?** The equation would penalize $\hat{a} - a = 0$ the same as $\hat{a} - a = \pm 180$ (i.e. both would return $L = 0$). So it would end up not penalizing a prediction that is the complete opposite orientation of the ground truth, and indeed would provide a gradient toward an even more incorrect answer if the initial prediction was more than 90 degrees off). Don't you need this output in order to “assign back and front to the region principal axis” [lines 386-387? How does the network learn from this loss except to randomly align with the axis prediction? Maybe I am missing something major because the network predictions are obviously good. In any case, it may be helpful to try out a different loss following Zhou et al. On the continuity of Rotation Representations in Neural Networks. CVPR 2019. DOI: 10.1109/CVPR.2019.00589.

We explain this point in the main comments of the Reviewer. We additionally paste it here for the ease of reading.

The loss function has a denominator equal 2. That means the range of loss is value of sine function between 0 and 180 deg where function values close to 0 result from orientation angle difference close to 0 and 360 degrees. Value close to 1 results from orientation angle difference close to 1. The square power of the sine function was missing in the text and we corrected this in the revised version of the manuscript.

In the implementation the orientation angles are scaled between 0 and 1. The angle values are multiplied by π (instead of 2π) before applying sine function which is equivalent to the function described in the text.

385: “axis full-bee” - should be “axis for full-bee”

The text has been corrected.

386: “a given regions” - should be “a given region”

We added these corrections to the text (l. 392, 395).

407: “We retrained the same approach...” - Do you mean you trained the network from Ref 13 and 35 from scratch on the rescaled images? “Retrained” makes it sound like the first training was part of the method for this paper (maybe it is?).

We actually trained this network from scratch on resized data. We reformulated this sentence in the revised manuscript (l. 417).

411: “We then retrained the detection model” - Do you mean fine-tuned? In general I have been a little confused in this section about what you mean when you say retrained.

Yes in this case we retrained. We simply used the pretrained network and fine-tuned it on a small dataset of corrected image annotations.

438-439: “We used both position coordinates as well as the posture categories...” - I think the authors may want to try also adding an orientation matching cost. If performance is still lower than when using visual features, it would serve as a powerful demonstration that the visual feature embedding is capturing something in addition to orientation. See main comments above.

We added the requested analyses and described them below the main comments of the Reviewer.

448: in the denominator inside the $\max(\cdot)$, is “ $i-1..n$ ” supposed to be “ $i=1..n$ ”?

Indeed this is a mistake which we have corrected. We thank the reviewer for their detailed reading.

449-450: “The length factor prioritizes longer trajectories” - this might be more clear if you rephrase to “prioritizes [adding new detections] to longer trajectories” (if this is correct)

We reformulated this sentence accordingly (l. 459).

455-456: “Unmatched trajectories are kept until the time between ... and the current time is below ...” - I think you mean “current time is above...”?

That was a mistake, we meant when the time exceeds the predefined gap and the text has been corrected.

481-482: “over the number of correct triplets with $L = 0$ ” - Where is this $L/\text{num_correct}$ coming from? Is this meant to encourage larger updates for batches where there were not very many correct triplets (i.e. hard batches)? What if there are no correct triplets in the batch?

Yes this a so called “batch all” strategy taking into account all triplets in a batch (see e.g. <https://omoindrot.github.io/triplet-loss>). We chose it over other strategies based on the time the network takes to converge. A small factor of $1 \cdot e^{-16}$ is added to the denominator to prevent division by zero.

490: “are fed back into training” - Do you mean they are added directly to the next batch?

Yes. The revised manuscript contains additional Fig. S11 illustrating this process.

495: “augmentation procedures ...” - Was each image within a triplet randomly rotated/flipped separately? I think **augmentation via a single randomly chosen rotation/flip that is then applied simultaneously to all three images in the triplet might improve performance** because it wouldn't introduce as much of a domain gap between training and augmented distributions. If relevant visual features are asymmetric, flipping images independently within a triplet could hurt performance. Similarly rotating the anchor and positive separately might also decrease performance if the main visual feature that is being extracted is the orientation of the bee. See comments for lines 227-228.

A single combination of augmentation operations was sampled for each triplet and applied to all three images. Modifying each image in a triplet in a different way would, to our understanding, introduce unnecessary variability in the data that does not exist in the actual data.

514: “ $D = BE + V + I$ ” - I think a few additional experiments replacing V with an orientation angle distance could really help demonstrate the value of matching using visual features over just orientation. See main comments above.

We performed the tests for orientation angle effect on performance and explained the results below the Reviewer's main comments.

522: “**manually validated**” - Please describe the procedure for manual validation including number of validators that scored each potential trajectory. For example, this could be an outlier, but

M16_tr1_P17_421_highincell_2764.17_4.00.mp4 contains an ID swap at $t = 57$ seconds. The target bee from preceding frames is occluded as several other bees walk over it and then continues to the left after the other bees walk over it. On the other hand, M17 shows a very impressive track recovery at $t = 1$ min 45 sec. I think these swaps have relatively little impact on overall training with the triplet loss, but their presence should be noted or at least estimated. In fact, these swaps could potentially be located by looking for spikes in D [line 514], calculated for each frame along a trajectory.

Indeed, this trajectory contains a swap however since it happens within the first minute of the recording and the remaining trajectory is correct it is still considered as a correct trajectory. A “correct” trajectory is defined as a temporally-contiguous set of registered positions in which the same organism was identified $\geq 80\%$ of the time span of a recording by manual inspection. Trajectories which are shorter than the recording length which correspond to bees that have entered or left the hive are considered correct as long as they track the bee throughout her entire presence in the hive. These clarifications have been added to the revised version of the manuscript l. 544-550, as well as in the legend of the Supplemental Table T3.

In training we use only segments from min 1 to 4 of the reference trajectories, this way ensuring these segments don't contain potential error allowed by our correctness criteria.

522-523: “**Correct trajectories**” - Please define what it means for a trajectory to be correct. From what I gather, it means a trajectory that is more than one minute long with no ID swaps?

We describe it in the point above and in the revised manuscript (l. 544-550).

529: “The network trained on this way expanded train set” - remove “way”

The text has been corrected.

530: “Images of video S5” - maybe should be “Images from video S5”?

The text has been corrected.

530: “were not used in training neither” - I don't think you want a double negative here. Maybe rephrase to “were not used in training either the detection or representation learning networks”?

We corrected the text as suggested in these three points.

590-591: (Figure 3E) “High numbers of cell-bees occur predominantly in the evening...” - Do you have any information whether these bees are “sleeping” or might they be feeding brood?

We don't yet know. As the current manuscript is already substantial (and focused mostly on tracking) we haven't performed a systematic recognition of the variety of observable bee behaviors.

A recent study (Slumber in a cell: honeycomb used by honey bees for food, brood, heating... and sleeping. Barrett A. Klein and M. Kathryn Busby. 2020) suggest this could be possible based on video data. We expect to expand this work with behavior detection and analysis in the future.

613: (Figure 4C) “an earlier approach” - This probably isn't feasible, but it would be great to a comparison with idtracker (refs 33, 34, 46).

We agree but unfortunately idTracker simply isn't yet feasible on such a dense collection of organisms. However, the idea that organism identity might still be contained, somehow, in visual appearance remains attractive and will be the basis of future work.

625: (Figure 5C) “The spatial distribution of trajectories characterized by large linear and angular motion.”
- What does color measure? The number of observations for each position bin?

We added explanation of the colors to this figure legend in the revised manuscript.

Minor asks: Add equation numbers throughout unless not consistent with journal formatting.

We added equation numbers to the revised manuscript.

Supplement

Figures S4 and S5. The meaning of black and gray lines for raw and smoothed values appears to have switched? It would be great if they were consistent. Also, please add a description of these to the figure legend.

We changed the coloring to match Fig. 3C (now Fig. S3-S4).

Figure S7. please consider adding supporting statistics using a circular analysis of variance (e.g. with the package 'circular' in R) including hive as a 'group'. Figure S9. change “bell-bee” to “cell-bee”

We included circular statistics in the revised manuscript as suggested by the Reviewer (now Fig. S6).

Figure S9. It could be more informative to have time shift range from -12 h to 12 h, which would make the high brood number plot more interpretable (i.e. the orange vertical line would be to the left of the green vertical line).

We changed the range as suggested by the Reviewer (now Fig. S8).

Supp Table 1: Why does Supp T1 not have an entry for human true positive rate? Is it 1.0? See my comment for lines 348-351. Also, the "Axis" column measurement is not defined anywhere in the manuscript. This is the error of the principal component direction of the network segmentation, correct? “Orientation angle”

would be the error after assigning head/tail based on mean orientation pixel values for the region? It could be helpful to add these definitions to the table legend.

We added estimates of human accuracy to Supplementary T1. As described above, we use a reference, double-checked data to estimate error of human labelers. We also removed the “Axis” column from this table – it pertains to the technical aspects of development of the method that we don’t expand on in this manuscript. “Orientation angle” indeed is the orientation assigned to the principal component based on the pixel orientation values.

T3 description, third line: Change “The trajectories constructed by each tracking methods are...” to “The trajectories constructed by each tracking method are...” (methods to singular).

We corrected the caption as suggested.

Reviewers' Comments:

Reviewer #2:

Remarks to the Author:

The manuscript is fine. I find the authors have provided satisfactory replies to my queries of more biological nature. I hope the upscaling to more realistic colony size will succeed.

Reviewer #3:

Remarks to the Author:

The authors have thoroughly addressed all of my comments. This work is novel and addresses an urgent need that is shared across biomechanics, entomology, and behavioral ecology communities. I think this manuscript is ready for publication.

I have one additional comment about individual re-identification (Re-ID) in this context. If individuals move in and out of the camera frame and long term analyses require individual-level data, Re-ID is obviously very important. However, there are many questions that can be answered with tracks that are consistent across medium-length time periods (where track length is primarily limited by the duration that individuals are in the camera frame). Thus, this manuscript is a valuable contribution even if it does not solve the "holy grail". This work tracks an order of magnitude more individuals than do idTracker [44] and idtracker.ai [46] and I question the ability of [44] and [46] to accurately re-identify individuals on the scale of 1000+ individuals. If the Re-ID task is even possible at this scale, it will likely require much higher resolution imaging and an embedding that is explicitly invariant to body orientation and pose, the presence/absence of pollen, and background. The method and dataset presented here, as the authors point out, are the necessary first steps toward developing such a system.

Sincerely,
Marc Badger

Reviewer #4:

Remarks to the Author:

Summary

The paper proposes a marker-less bee tracking system, and I agree with the authors that this is an interesting and promising research direction with significant potential in its applications. Generally, I agree with reviewer 1's assessment of the paper and even though the authors have improved the manuscript by integrating the reviewers' feedback, it still seems quite raw - both conceptually and composition-wise. After reading the revised version of the manuscript I am still left with the feeling that this could be a much better and cleaner paper.

The described method is an incremental improvement to a previously published method (in terms of implementation details and tracking performance). That in itself can and should not preclude publication but the text appears cluttered with seemingly irrelevant information in some parts and missing details in others. Take the results section for example, it ends with several paragraphs with descriptions that do not represent novel findings relating to bee biology, nor are these results framed and introduced as specific evaluations to describe the performance of the method. To be honest, I didn't enjoy reading the manuscript because it isn't a proper method paper (which I think it should be), nor is it a study investigating specific hypotheses (which seems possible given the impressive datasets that the authors recorded).

The method can be of interest to biologists and the modeling community but the manuscript fails to describe specific use-cases that stand out against marker-based tracking systems. The authors

stated that the method can be used in animals that can't be tagged but I think there are more direct applications. I would have enjoyed the introduction much more if the authors didn't list disadvantages of marker-based systems, some of which seem to have been exaggerated (see below in my detailed response). It would have been better to describe use cases in which marker-less tracking systems are simply the better choice and knowing the identities of your animals isn't required. For example, if we wanted to assess colony-level descriptions that integrate individual-level dynamics (e.g. distribution of motion speeds) to quantify the effects of pesticides or illnesses and pests, we probably wouldn't need individual IDs. It would be much more credible if you would sell this not as a replacement for marker-based systems, or if you make it sound as if this just needs a few more years and then we can re-identify foragers that left the hive. The latter is just misleading and there is no evidence that would suggest this could work.

Also, the brood detection module is a nice proof of concept but others have already shown this to work with even more output classes. It would now seem prudent to put this method to use to investigate a research question. You combined brood and bee detection and showed a correlation between the number of cell-bees and brood cells but this a) isn't novel and b) nor was it introduced as a research question nor was it framed as a method to evaluate your system.

In summary, I do not think that the manuscript in its current form is suitable for publication in Nature Communications. See below for detailed feedback.

Feedback to response to reviewer's comments (line numbers refer to respective reviewer comments in the rebuttal document)

184: It's still unclear how your method compares to other systems. Runtime performance: you mention the supercomputer used in Wario et al (2015). I think this info is outdated and should be compared to other works using GPU-based computing. After a quick search: Wild et al (2018) reported 2 ms per bee and frame.

208: Tracking evaluation: You refer to lines 544-550, but this is the description of the training data. You write you counted how many trajectories contained $\geq 80\%$ of the same animal. Does this mean that if you started with animal A, then one frame later the track jumped to and stayed on animal B, that this was counted as a correct track?

228: Regarding background masking and artifacts: you don't mention any artifacts in 229-231.

234: One of the most important points reviewer 1 raised: The tracking evaluation remains rather simple, with many metrics not being shown. Why do you count tracks with 20% errors as correct? I think it would make sense to define typical errors (missing detections in a track, trajectory jumps to another animal, etc) and then show their distribution. It would also be very interesting to see over which distances your method is able to assign the correct correspondences. I would have liked to see an analysis in which you downsample video frame rate and evaluate tracking performance. You evaluate on a dataset from a separate colony (good!) but why don't you evaluate on recordings from the other colonies too (per-colony cross validation). Would be good to show some characteristics of those colonies, like distribution of motion speeds, etc, to have an idea of how "hard" this test was. In terms of evaluation I would have liked to see much more "stress tests", or visualizations that show under which conditions the tracking breaks.

25: Dancer/follower detection: the rank sum test doesn't provide a useful metric like classification error or accuracy would.

26: You say that the tracking reveals behaviors like waggle dancing or cell inspections but do not show evaluations. The correct integration of cell-inspecting bees wasn't evaluated.

132: Circular distribution is really hard to read (multiple overlaps)

133: How was the threshold 800 determined (which percentile?)

419: The description of the background extraction method lacks clarity.

Feedback to Manuscript

22: "nightly enhancement of cell-bees" - Before the definition of cell-bees I didn't understand this.

38-39: "a full quantitative understanding of the colony behavior" - I would love to see here a (short) list of what exactly this could mean in the future. Either add this here or later where you introduced marker-based systems.

42: A "natural honey bee colony" may be misleading the reader to think of colonies in hollow trees, etc.

45: Citations missing: Mersch et al 2013 (the first marker-based insect tracking system) and Blut et al. 2017 (the same system used in honeybees)

47: newer citations missing: e.g. Geffrea et al. 2020

49-57: This paragraph should be rewritten to make the case for marker-less tracking systems not against marker-based systems.

51: "Recognition of markers is hampered by visual occlusions" - Losing an animal due to occlusions can happen with or without markers. I would even say that marker-based systems can recover much easier.

54: "limiting behavioral repertoires" - Again, I wouldn't present this as a binary choice, that either you use markers or not. Some groups who use markers also use computer vision algorithms to extract additional information (e.g. to detect food exchange behaviors in Gernat et al. 2018). I can easily see a combination of markers and marker-less methods becoming the method of choice in the future.

55: "physical tags cannot be used to identify brood, honey, or pollen storage" - Similar argument as above. You shouldn't position cell classification as a point against marker-based systems. It's a different task and even your method uses two different convnets. Can't someone who needs marked bees use your second network to classify brood cells?

58 - 67 This paragraph is trying to make the point that there hasn't been much work on multi-object tracking in bees. Generally, I'd say that is true but you are missing a few citations (Poiesi & Cavallaro, 2015, Kimura et al. 201X) and I would love to read why methods that can track human crowds can't be used in honey bees? I am not too familiar with this field but a quick search tells me there are solutions (e.g. Kothari et al 2020). The fact that bees live in dense groups as an argument for novel systems isn't really convincing. Using a sufficiently high framerate and/or limiting the number of potential candidates from within the immediate vicinity of the object makes the task easier (your method does that too) and existing methods could be used. Tracking 2000 bees isn't really that much different from tracking 200 humans (ignoring for a moment the computational costs). Besides, over the last years many animal tracking systems have been introduced, some of which say they can cope with multi-object scenarios. Why can't we use those existing methods (e.g. DeepLabCut)? I am not saying that we should or that I have an answer to this but I think many readers will ask that question.

73: I believe this is inaccurate. A segment needs to have no overlaps only for one pair of fish, not

for all 100 fish (not really sure here).

75: The idea of extracting (learned) image features that inform the tracking step is not new and should be put into context (see e.g. Kalal et al. 2010).

122-125: It seems like you use two separate networks for bee and comb detection. Why not use additional output channels?

130-131: "in a standardized imaging setup and based on static images" - It is unclear to me what "standardized imaging setup" means (to a degree you seem to have standardized your setup as well) and why static images are different from what you do (I understand your segmentation network uses only a single frame).

151: "toxins in the environment" - Speculation. I would remove this part.

178-179: It remains unclear whether you wouldn't need individual identification for this. Anyhow, this is exactly one of those sentences you could put in the introduction to motivate marker-less tracking systems.

185-187 This refers to a concept called "contrastive learning" and should be put in correct context.

187-189 I don't see why anyone would even want to train on all possible triplets. It's the likely ones that you want to be able to discriminate.

229-231: The background plays an important role in correctly identifying the animals. It would be great to see how sensitive this is over longer spatial distances.

242 - 276: Frankly, I don't see how these analyses tell us anything about the validity of the tracking system or the biology of the colonies. See also my general comments above.

549-550: "The proportion of correctly tracked bees is quantified relative to the mean number of bee detections in a recording." - I don't understand this. You first said you inspect trajectories and count them as correct if 80% of the detections were from the same animal (by the way, this could mean an early jump of the trajectory and still a correct track, right?). Now you say the "proportion of correctly tracked bees" does this refer to all detections in a correct trajectory?

We thank the reviewer for the comments which have helped us draft a better manuscript.

Reviewer #4 (Remarks to the Author):

Summary

The paper proposes a marker-less bee tracking system, and I agree with the authors that this is an interesting and promising research direction with significant potential in its applications. Generally, I agree with reviewer 1's assessment of the paper and even though the authors have improved the manuscript by integrating the reviewers' feedback, it still seems quite raw - both conceptually and composition-wise. After reading the revised version of the manuscript I am still left with the feeling that this could be a much better and cleaner paper.

The described method is an incremental improvement to a previously published method (in terms of implementation details and tracking performance). That in itself can and should not preclude publication but the text appears cluttered with seemingly irrelevant information in some parts and missing details in others. Take the results section for example, it ends with several paragraphs with descriptions that do not represent novel findings relating to bee biology, nor are these results framed and introduced as specific evaluations to describe the performance of the method. To be honest, I didn't enjoy reading the manuscript because it isn't a proper method paper (which I think it should be), nor is it a study investigating specific hypotheses (which seems possible given the impressive datasets that the authors recorded).

The method can be of interest to biologists and the modeling community but the manuscript fails to describe specific use-cases that stand out against marker-based tracking systems. The authors stated that the method can be used in animals that can't be tagged but I think there are more direct applications. I would have enjoyed the introduction much more if the authors didn't list disadvantages of marker-based systems, some of which seem to have been exaggerated (see below in my detailed response). It would have been better to describe use cases in which marker-less tracking systems are simply the better choice and knowing the identities of your animals isn't required. For example, if we wanted to assess colony-level descriptions that integrate individual-level dynamics (e.g. distribution of motion speeds) to quantify the effects of pesticides or illnesses and pests, we probably wouldn't need individual IDs. It would be much more credible if you would sell this not as a replacement for marker-based systems, or if you make it sound as if this just needs a few more years and then we can re-identify foragers that left the hive. The latter is just misleading and there is no evidence that would suggest this could work.

Also, the brood detection module is a nice proof of concept but others have already shown this to work with even more output classes. It would now seem prudent to put this method to use to investigate a research question. You combined brood and bee detection and showed a correlation between the number of cell-bees and brood cells but this a) isn't novel and b) nor was it introduced as a research question nor was it framed as a method to evaluate your system.

In summary, I do not think that the manuscript in its current form is suitable for publication in Nature Communications. See below for detailed feedback.

We understand from the comments that our manuscript is not exactly the manuscript that the reviewer would have written, but we appreciate the constructive feedback and hope that our detailed changes and additional analyses have clarified and strengthened the report of our primary advance: we do have the first dense object tracking algorithm that demonstrably

works in an observation honey bee hive. We would also like to address some important misunderstandings. (1) Our approach is NOT an incremental improvement. There is simply no published markerless tracking of an insect colony. In our own earlier work, we did explore a ‘pixel identity’ approach, an approach that ultimately proved unworkable but from which we gained applicable experience. The method presented in this current manuscript is an entirely new solution to the honey bee tracking problem. We have revised our wording to clarify this point (l. 85). (2) The detection and tracking performance are evaluated within the scope of what is feasible to label in a first dataset which encompasses millions of detections and thousands of trajectories. We thank the reviewer for the suggestion to add stress and cross-validation test analyses and have done so (l. 236-237, 243-247 new Supplemental Figures S18, S19, S20). (3) Our analyses are intended to demonstrate the possibilities with the tracked data, both with the brood dynamics and with the bee trajectories. In particular, we sought to demonstrate that even with ~5 min trajectories we find plenty of interpretable and ethologically relevant behaviors, regardless of their novelty.

By listing the advantages of markerless tracking we aimed to motivate our approach not dismiss the importance of marker-based systems and indeed we believe that the techniques are very much complementary. We make this point more explicit in the revised version of the manuscript (l. 49-59, 331-332). We do note that the ability to easily track multiple hives and to capture posture information such as crawling inside a comb cell is unique to markerless approaches.

We respectfully disagree with the reviewer’s contention that re-identification of foragers as they return to the hive will not be possible in the coming years. We are not naive about the challenge and indeed our “pixel personality” work convinced us of the inadequacy of current methods in tracking organism identity from machine vision in dense, natural environments. But we are informatively optimistic given the pace of development in computer vision, the growing capacity of machine learning methods to detect visual patterns imperceptible to human eye, and the ability to directly tailor imaging systems to the computational task. However, this point is not important for the current manuscript and we have removed the statement from the discussion (l. 318-320).

Finally, although no exact reference was given in the referee report for previous work on brood detection, we assume that the referee is referring to Alves et al (reference 56 in our manuscript). While this study does detect more types of cell content than ours, its application is on empty frames that were taken out of the hive and put into an ‘imaging tunnel’ to obtain standard and good quality image. In our setup there is noise introduced by the active bee colony living on the frame. In contrast to Alves et al., we thus obtain continuous information about the brood count in a live colony without disrupting it. We have added these clarifications to the revised manuscript (l. 134-137).

Feedback to response to reviewer’s comments (line numbers refer to respective reviewer comments in the rebuttal document)

184: It’s still unclear how your method compares to other systems. Runtime performance: you

mention the supercomputer used in Wario et al (2015). I think this info is outdated and should be compared to other works using GPU-based computing. After a quick search: Wild et al (2018) reported 2 ms per bee and frame.

We thank the Reviewer for noticing this omission. 2 ms-long tag recognition time per bee in a beehive of ~1,000 bees would amount to 1 h 40 min for a 5 min recording at 10 FPS that we used in our study (1 h on a GPU node used by us). Notably in Wild et al. 2 ms does not include tag detection time which is performed by another neural network the computational cost of which is not provided. The time estimation of Wild et al. is still on a Cray X30 supercomputer with 1200 CPUs which again makes the comparison difficult.

208: Tracking evaluation: You refer to lines 544-550, but this is the description of the training data. You write you counted how many trajectories contained $\geq 80\%$ of the same animal. Does this mean that if you started with animal A, then one frame later the track jumped to and stayed on animal B, that this was counted as a correct track?

Reviewer 1 asked here about trajectory validation method and our response and text in 544-550 clarifies this. The trajectories were validated based on visual inspection and we did allow for their incompleteness in either first or last minute of the recording. The correct parts of each trajectory between minutes 1 and 4 of the recording are next used in network training.

228: Regarding background masking and artifacts: you don't mention any artifacts in 229-231.

We rephrased this sentence (l. 236).

234: One of the most important points reviewer 1 raised: The tracking evaluation remains rather simple, with many metrics not being shown. Why do you count tracks with 20% errors as correct? I think it would make sense to define typical errors (missing detections in a track, trajectory jumps to another animal, etc) and then show their distribution. It would also be very interesting to see over which distances your method is able to assign the correct correspondences. I would have liked to see an analysis in which you downsample video frame rate and evaluate tracking performance. You evaluate on a dataset from a separate colony (good!) but why don't you evaluate on recordings from the other colonies too (per-colony cross validation). Would be good to show some characteristics of those colonies, like distribution of motion speeds, etc, to have an idea of how "hard" this test was. In terms of evaluation I would have liked to see much more "stress tests", or visualizations that show under which conditions the tracking breaks.

We take the 80% correctness as a metric broadly used in computer vision community for objects regarded as "mostly tracked" (Leal-Taixé, 2016, Li et al. 2009, Milan et al. 2016, and many more). Given this choice of trajectory correctness, we show in new Fig. S18 a computation of the distribution of distances to the next detection for identity swaps, lost tracks and correct trajectories. The identity swaps and lost trajectories are much more likely to have larger distances to the next detection.

To add further trajectory quantification, we performed the distance stress test as suggested by the Reviewer. We temporally down-sampled the recordings by a factor of 2 and 10 resulting in videos of 5 FPS and 1 FPS, respectively, and investigated the occurrence of identity swaps and trajectory losses with respect to bee density around the tracked bee and the distance of the next position in the track. We find that the distance of the next position in the trajectory has a

very strong effect on the tracking errors – with larger distances more errors occur. The effect of the density of bees around the track is less striking. We added the new analysis as new Supplemental Figures S18 and S19 and refer them in the manuscript (l. 236-238).

We additionally performed the tracking cross-validation test. We trained the network from scratch four times, each time leaving out one of the hives (S1-S4). The embeddings were next calculated in the left-out recording and trajectories were constructed. The results, together with the originally presented results in S5 are shown in the new Supplemental Fig. S20. The proportion of tracked bees does not differ greatly across the hives and is lowest in the hives with large proportion of fast-moving bees (S3, S4). We provide colony characteristics (distributions of speed, angular speed, area of motion, diffusion coefficient) in Fig. 5A-B and Fig. S22 (provided in the original version of the manuscript) and describe the performed test in the revised manuscript (l. 244-247).

25: Dancer/follower detection: the rank sum test doesn't provide a useful metric like classification error or accuracy would.

Here, we are not proposing recognition method but simply demonstrating how motion parameters collected from our method can help to identify behavior. The striking overrepresentation of certain behaviors in our rankings provides a strong foundation for building behavior recognition methods in the future. We added this explanation to the revised manuscript (l. 286).

26: You say that the tracking reveals behaviors like waggle dancing or cell inspections but do not show evaluations. The correct integration of cell-inspecting bees wasn't evaluated.

Here as well as in the point above, we only point out what behaviors can we have and easier access to via simple metric filtering. Our focus here is on tracking and not on recognizing behavior.

132: Circular distribution is really hard to read (multiple overlaps)

It is exactly the overlap of daily activity that is key in this plot. The lack or presence of non-overlapping parts of the distribution carries the information that we want to get across with this plot.

133: How was the threshold 800 determined (which percentile?)

We found 800 as the highest number of brood that was observed across almost all of the hives

419: The description of the background extraction method lacks clarity.

We reformulated this description (l. 443-445).

Feedback to Manuscript

22: "nightly enhancement of cell-bees" - Before the definition of cell-bees I didn't understand this.

We changed this formulation (l. 22).

38-39: “a full quantitative understanding of the colony behavior” - I would love to see here a (short) list of what exactly this could mean in the future. Either add this here or later where you introduced marker-based systems.

We consider automated recognition of bee behavior as an important part of quantitative picture of bee colony behavior. We added this sentence to the manuscript (l. 47-49).

42: A “natural honey bee colony” may be misleading the reader to think of colonies in hollow trees, etc.

We removed this term (l. 42).

45: Citations missing: Mersch et al 2013 (the first marker-based insect tracking system) and Blut et al. 2017 (the same system used in honeybees)

We added these citations.

47: newer citations missing: e.g. Geffrea et al. 2020

We added this citation.

49-57: This paragraph should be rewritten to make the case for marker-less tracking systems not against marker-based systems.

We rewrote this paragraph to tone it down (l. 49-59).

51: “Recognition of markers is hampered by visual occlusions” - Losing an animal due to occlusions can happen with or without markers. I would even say that marker-based systems can recover much easier.

Partial occlusion of a marker can hamper its recognition as the barcodes are very information dense. Whereas partially occluded bees are still detected in our approach and are tracked as depicted by the tracks of bees coming in and out of the cells. We rewrote this sentence (l. 53) accordingly.

54: “limiting behavioral repertoires” - Again, I wouldn’t present this as a binary choice, that either you use markers or not. Some groups who use markers also use computer vision algorithms to extract additional information (e.g. to detect food exchange behaviors in Gernat et al. 2018). I can easily see a combination of markers and marker-less methods becoming the method of choice in the future.

We do not present this as a binary choice in this sentence. We state that “crawling inside of a honeycomb cell or walking upside down on the glass of the observation hive” is not detectable with the use of tags, which is true.

55: “physical tags cannot be used to identify brood, honey, or pollen storage” - Similar argument as above. You shouldn’t position cell classification as a point against marker-based systems. It’s a different task and even your method uses two different convnets. Can’t someone who needs marked bees use your second network to classify brood cells?

We removed this part of the sentence (l. 57).

58 - 67 This paragraph is trying to make the point that there hasn’t been much work on multi-object tracking in bees. Generally, I’d say that is true but you are missing a few citations (Poiesi & Cavallaro, 2015, Kimura et al. 201X) and I would love to read why methods that can track human

crowds can't be used in honey bees? I am not too familiar with this field but a quick search tells me there are solutions (e.g. Kothari et al 2020). The fact that bees live in dense groups as an argument for novel systems isn't really convincing. Using a sufficiently high framerate and/or limiting the number of potential candidates from within the immediate vicinity of the object makes the task easier (your method does that too) and existing methods could be used. Tracking 2000 bees isn't really that much different from tracking 200 humans (ignoring for a moment the computational costs). Besides, over the last years many animal tracking systems have been introduced, some of which say they can cope with multi-object scenarios. Why can't we use those existing methods (e.g. DeepLabCut)? I am not saying that we should or that I have an answer to this but I think many readers will ask that question.

Applying tracking methods designed for humans on bee data will not work for several reasons. Most importantly humans are much more visually distinct from one another. Moreover, the dynamics of their motion is dramatically different from the irregular motion of the bees. Also, the lack of labeled datasets with bee trajectories is a major obstacle to developing such methods. We respectfully disagree with the Reviewer that there is no difference between tracking a set of 2000 bee and 200 human individuals. The difficulty of the problem grows exponentially with the groups size as the density and the number of potential identity swaps becomes much larger. Automatically detecting visual features that are distinct between tracked individuals becomes massively more difficult in a much larger group of identical objects.

Finally, DeepLabCut and other recent pose estimation algorithms such as Leap and DeepPoseKit are designed for tracking body parts of individual animals, at higher resolution than our bee recordings. While insect "pose" certainly carries substantial interesting information, its estimation has not been achieved for dense groups. There are also recent advances for pose estimation with multiple organisms (SLEAP, DeepLabCut) but these apply to a small number of organisms in a non-dense setting.

73: I believe this is inaccurate. A segment needs to have no overlaps only for one pair of fish, not for all 100 fish (not really sure here).

All fish need to be non-overlapping, the initial step of the algorithm consists of detecting 'global fragments' and choosing the most appropriate one to base the tracking algorithm on.

75: The idea of extracting (learned) image features that inform the tracking step is not new and should be put into context (see e.g. Kalal et al. 2010).

We were not familiar with this author and added his work to the revised manuscript (l. 73).

122-125: It seems like you use two separate networks for bee and comb detection. Why not use additional output channels?

Comb detection is applied on background extracted images, it requires prior background extraction, there are therefore no bees in those images.

130-131: "in a standardized imaging setup and based on static images" - It is unclear to me what "standardized imaging setup" means (to a degree you seem to have standardized your setup as well)

and why static images are different from what you do (I understand your segmentation network uses only a single frame).

By static we mean, there are no bees in those images, we rephrased this sentence (l. 134).

151: “toxins in the environment” - Speculation. I would remove this part.

It is a hypothesis that is supported by previous studies that we cite in this sentence, we decided to keep this sentence.

178-179: It remains unclear whether you wouldn't need individual identification for this. Anyhow, this is exactly one of those sentences you could put in the introduction to motivate marker-less tracking systems.

We now added information about the time span of recordings to the introduction (l. 58).

185-187 This refers to a concept called “contrastive learning” and should be put in correct context.

We believe we cite the relevant literature in this part.

187-189 I don't see why anyone would even want to train on all possible triplets. It's the likely ones that you want to be able to discriminate.

We agree. Indeed, in the following sentences we describe our strategy to identify the likely triplets.

229-231: The background plays an important role in correctly identifying the animals. It would be great to see how sensitive this is over longer spatial distances.

We performed the tests as suggested by the Reviewer (point referring to l. 234 above)

242 - 276: Frankly, I don't see how these analyses tell us anything about the validity of the tracking system or the biology of the colonies. See also my general comments above.

We appreciate the Reviewer's skepticism. As explained below the main comment of the Reviewer – these are examples meant to simply demonstrate the potential of our method. Our primary aim is in tracking which we hope will facilitate discoveries of bee biology.

549-550: “The proportion of correctly tracked bees is quantified relative to the mean number of bee detections in a recording.” - I don't understand this. You first said you inspect trajectories and count them as correct if 80% of the detections were from the same animal (by the way, this could mean an early jump of the trajectory and still a correct track, right?). Now you say the “proportion of correctly tracked bees” does this refer to all detections in a correct trajectory?

We define “correct trajectory” two lines prior (l. 557-559). By correctly tracked we mean correct trajectories.

Reviewers' Comments:

Reviewer #4:

Remarks to the Author:

One of my main concerns about the paper was its lack of focus on the method. In my eyes, the descriptions in the results section are still rather underwhelming (neither novel nor systematic) and I, reading the ms again, asked myself why these results appear here. I would rather have enjoyed a thorough analysis of what your tool can do and in which situations it fails.

Fortunately, the authors have included an analysis of their method's performance in the SI (see below for remaining questions) and I think this will increase the readers' trust and understanding. I think, overall this is impressive work and represents an important step towards a comprehensive (computational) understanding of honeybee collective behavior. I recommend acceptance for publication.

Remaining questions:

- Defining a correct track to be > 80% correct still feels weird and the reader will ask why it was defined that way. It makes it harder to understand your results and feels like unnecessary obfuscation. I still think it would be better to have a set of metrics, each focusing on errors that could happen, or applications that readers may have in mind: A track could be incorrect because of ID swaps (bad for some applications) or just missing detections, can the system recover from occlusions of duration X? If I need a track at least X min long, how many of these can your system provide?
- You said in your rebuttal that only minutes 1-4 (60% of the track) are used in the validation of the trajectory. Why? And does this relate to the 80% threshold above (if so, how)?
- S18, S19: what is a "lost" trajectory? How is this different from an ID swap? S19 shows that reducing the framerate to 5 fps still provides almost the same performance (meaning you can save 50% of storage or record twice as long)! Why not report this in the main text?

We thank the reviewer for the positive evaluation of the manuscript and for the final comments that we address here below.

Remaining questions:

Defining a correct track to be $> 80\%$ correct still feels weird and the reader will ask why it was defined that way. It makes it harder to understand your results and feels like unnecessary obfuscation. I still think it would be better to have a set of metrics, each focusing on errors that could happen, or applications that readers may have in mind: A track could be incorrect because of ID swaps (bad for some applications) or just missing detections, can the system recover from occlusions of duration X ? If I need a track at least X min long, how many of these can your system provide?

Our detection method is fairly accurate (l. 119-121, Fig. S1, Table T1), therefore ID swaps and trajectory loss (trajectory ending inside the hive before the end of the recording) are the major sources of errors. To answer the first question, we performed an occlusion test. We sampled 100 trajectories from each recording at random and introduced gaps of increasing size to the sampled trajectories, one trajectory at a time. We next performed a matching step with detections in the frame following the gap – the sampled trajectory including a gap and all remaining trajectories being complete. While our tracking is robust to short gaps, for occlusions longer than ~ 2.5 sec we can no longer correctly track across $\sim 50\%$ of the trajectories, and we include these results in an additional panel of Supplementary Figure 19. In future work we believe that we could substantially improve this performance by explicitly including gap challenges in the training.

To answer the second question, we generated cumulative plot of the trajectory durations in all five recordings.

Most of the trajectories span the entire 5 min of the recordings. This result is shown in more detail in the Supplementary Figure 21.

You said in your rebuttal that only minutes 1-4 (60% of the track) are used in the validation of the trajectory. Why? And does this relate to the 80% threshold above (if so, how)?

The 80% correctness criterion results in trajectories that span without errors at least 4 min. This gives us certainty that in a 5 min recording errors might occur either in the first or last

minute of the recording and no error is present in the correct trajectories between minutes 1 and 4. We added this clarification to the revised manuscript (l. 572-573).

S18, S19: what is a "lost" trajectory? How is this different from an ID swap? S19 shows that reducing the framerate to 5 fps still provides almost the same performance (meaning you can save 50% of storage or record twice as long)! Why not report this in the main text?

“Lost” trajectory is one that ends inside the hive before the end of the recording, we added this clarification to the caption of Supplementary Fig. 18. Due to the scales in Supplementary Fig. 19 the differences between 5 FPS and 10 FPS can be misinterpreted – error numbers increase 0.3-2.5 times between 5 FPS and 10 FPS and 4-40 times between 1 FPS and FPS. We added this clarification to the image caption.